# Class-attribute Priors: Adapting Optimization to Heterogeneity and Fairness Objective

**Anonymous**[1]

[1]Anonymous Institution

**Abstract** Modern classification problems exhibit heterogeneities across individual classes: Each class may have unique attributes, such as sample size, label quality, or predictability (easy vs difficult), and variable importance at test-time. Without care, these heterogeneities impede the learning process, most notably, when optimizing fairness objectives. We propose an effective and general method to personalize the optimization strategy of individual classes so that optimization better adapts to heterogeneities. Concretely, class-attribute priors (CAP) is a meta-strategy which generates a class-specific strategy based on attributes of that class. This meta approach leads to substantial improvements over naive approach of assigning separate hyperparameters for each class. We instantiate CAP for loss function design and posthoc logit adjustment, with an emphasis on label-imbalanced problems. We show that CAP is competitive with prior art and its flexibility unlocks noticeable improvements for fairness objectives beyond balanced accuracy. Finally, we evaluate CAP on problems with label noise as well as weighted test objectives to showcase how CAP can jointly adapt to different types of heterogeneities.

## 1 Introduction

Contemporary machine learning problems arising in natural language processing and computer vision often involve large number of classes to predict. Collecting high-quality training datasets for all of these classes is not always possible, and realistic datasets [25, 10, 11] suffer from class-imbalances, missing or noisy labels (among other application-specific considerations). Optimizing desired accuracy objectives with such heterogeneities poses a significant challenge and motivates the contemporary research on imbalanced classification, fairness, and weak-supervision. Additionally, besides distributional heterogeneities, we might have objective heterogeneity. For instance, the target test accuracy may be a particular weighted combination of individual classes, where important classes are upweighted.

A plausible approach to address these distributional and objective heterogeneities is designing optimization strategies that are tailored to individual classes. To this end, arguably the simplest approach is assigning individual weights to classes during optimization. The recent proposals on imbalanced classification [23, 4] can be viewed as generalization of weighting and can be interpreted as developing unique loss functions for individual classes. More generally, one can use class-specific data augmentation schemes, regularization or even optimizers (e.g. Adam, SGD, etc) to improve target test objective. While promising, this approach suffers when there are a large number of classes $K$: naively learning class-specific strategies would require $\mathcal{O}(K)$ hyperparameters ($\mathcal{O}(1)$ strategy hyperparameter per class). This not only creates computational bottlenecks but also raises concerns of overfitting for tail classes with small sample size.

To overcome such bottlenecks, we introduce the **Class-attribute Priors (CAP)** approach. Rather than treating hyperparameters as free variables, CAP is a meta-approach that treats them as a function of the class attributes. As we discuss later, example attributes $\mathcal{A}$ of a class include its frequency, label-noise level, training difficulty, similarity to other classes, test-time importance, and more. Our primary goal with CAP is building an **attribute-to-hyperparameter** function **A2H** that

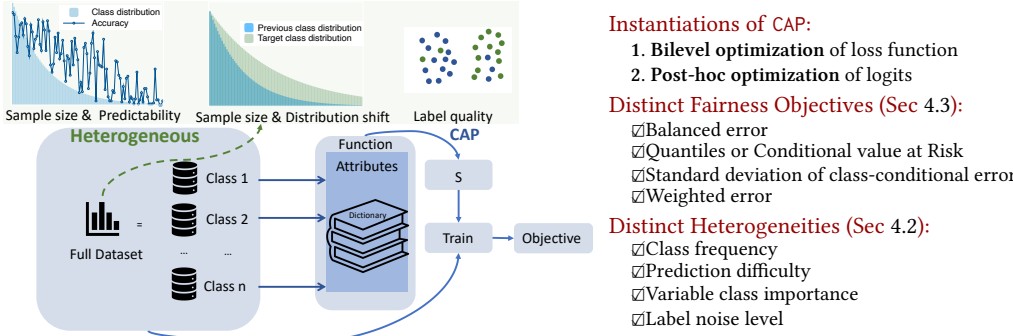

Figure 1: **Left hand side**: CAP views the global dataset as a composition of heterogeneous sub-datasets induced by classes. We extract high-level attributes from these classes and use these attributes to generate class-specific optimization strategies (which correspond to hyperparameters). Our proposal is efficiently generating these hyperparameters based on class-attributes through a meta-strategy. **Right hand side**: We demonstrate that CAP leads to state-of-the-art strategies for loss function design and post-hoc optimization. CAP can leverage multiple attributes to flexibly optimize a variety of test objectives under heterogeneities.

generates class-specific hyperparameters based on the attributes associated with that class. This process infuses high-level information about the dataset to accelerate the design of class-specific strategies. The **A2H** maps the attributes $\mathcal{A}$ to a class-specific strategy $\mathcal{S}$. The primary advantage is robustness and sample efficiency of **A2H**, as it requires $\mathcal{O}(1)$ hyperparameters to generate $\mathcal{O}(K)$ strategies. The main contribution of this work is proposing CAP framework and instantiating it for loss function design and post-hoc optimization which reveals its empirical benefits. Specifically, we make the following contributions:

1. **Introducing Class-attribute Priors (Sec 3)**. CAP is a meta approach that utilizes the high-level attributes of individual classes to personalize the optimization process. Importantly, CAP is particularly favorable to tail classes which contain too few examples to optimize individually.

2. **Incorporating CAP improves existing approaches (Sec 4)**. By integrating CAP with existing label-imbalanced training methods, CAP not only improves their performance but also increases their stability, notably, AutoBalance [23] and logit-adjustment loss [25].

3. **CAP adapts to fairness objective (Sec 4.2)**. CAP's flexibility is particularly powerful for non-standard settings that prior works do not account for: CAP achieves significant improvement when optimizing fairness objectives other than balanced accuracy, such as standard deviation, quantile errors, or Conditional Value at Risk (CVaR).

4. **CAP adapts to class heterogeneities (Sec 4.3)**. CAP can also effortlessly combine multiple attributes (such as frequency, noise, class importance) to boost accuracy by adapting to problem heterogeneity.

Finally, while we instantiate CAP for the problems of loss-function design and post-hoc optimization, CAP-style meta-optimization approaches can have far-reaching consequences to the design of optimal augmentations, regularization, and optimizers. This work makes key contributions to fairness and heterogeneous learning problems in terms of methodology, as well as practical impact. An overview of our approach is shown in Fig. 1

## 1.1 Related Work

The existing literature establishes a series of algorithms, including sample weighting [21, 35, 4], post-hoc tuning [25, 41, 17, 15, 38], and loss functions tuning [3, 19, 25, 16, 6, 33, 42], etc. This work aims to establish a principled approach for designing a loss function for imbalanced datasets. Traditionally, a Bayes-consistent loss function such as weighted cross-entropy [37, 28] has been used. However, recent work shows it only adds marginal benefit to the over-parameterized model due to overfitting during training. [25, 38, 19] propose a family of loss functions formulated as

$\ell(y, f(x)) = \log\left(1 + \sum_{k \neq y} e^{l_k - l_y} \cdot e^{\Delta_k f_k(x) - \Delta_y f_y(x)}\right)$ with theoretical insights, where $f(x)$ denotes the output logits of $x$ and $f_y(x)$ represents the entry that corresponds to label $y$. Above methods determine the value of $l$ and $\Delta$ to re-weight the loss function so the optimization generates a class-balanced model. In addition to these methods, [23] proposes a bilevel training scheme that directly optimizes $l$ and $\Delta$ on a sufficient small imbalanced validation data without the prior theoretical insights. However, the theory-based methods require expertise and trial and error to tune one temperature variable, making it time-consuming and challenging to achieve a fine-grained loss function that carefully handles each class individually. Although the bilevel-based method consider each class separately and personalizes the weight using validation data, optimizing the bilevel problem is typically time-consuming due to the computation of the Hessian-vector-product. Bilevel optimization is also brittle, especially when [23] optimizes the inner loss function, which continually changes the inner optima during the training.

Concerning the general goal, which is to ensure fairness with respect to protected target classes, several suggestions have been made in the literature [32, 20]. Balanced error and standard deviation [2, 1] between subgroup predictions are widely used metrics. However, they are insensitive to certain types of imbalances. The Difference of Equal Opportunity (DEO) [19, 11] was proposed to measure true positive rates across groups. [39] focus on disparate mistreatment in both false positive rate and false negative rate. Many modern machine learning tasks require models with high tail performance, focusing on certain underrepresented groups that normal machine learning models often neglect. Recent work has designed techniques for learning models with high tail performance instead of merely performing well on average [12, 30, 31, 23, 18]. The worst-case subgroup error is commonly used in recent papers [19, 30, 31]. Another popular metric to evaluate the model's tail performance is the CVaR (Conditional Value at Risk) [36, 40, 26], which computes the average error over the tails. Previous works [12, 8, 14, 26, 22] also measure tail behaviour using Distributionally Robust Optimization (DRO).

## 2 Problem Setup

This paper investigates the advantages of utilizing attribute-based personalized training approaches for addressing heterogeneous classes in the context of class imbalance, label noise, and fairness objective problems. We begin by presenting the general framework, followed by an examination of specific fairness issues, which encompass both distributional and objective heterogeneities.

Consider a multi-class classification problem for a dataset $(x_i, y_i)_{i=1}^N$ sampled i.i.d from a distribution with input space $\mathcal{X}$ and $K$ classes. Let $[K]$ denote the set $\{1..K\}$ and for the training sample $(x, y)$, $x \in \mathcal{X}$ is the input and $y \in [K]$ is the output. $f : \mathcal{X} \to \mathbb{R}^K$ represents the model and $o$ is the output logits. $\hat{y}_{f(x)} = \arg\max_{k \in [K]} o_k$ is the predicted label of the model $f(x)$. We also denote $K \times K$ identity matrix by $I_K$. Moreover, in the post-hoc setup, a logit adjustment function $g : \mathbb{R}^K \to \mathbb{R}^K$ is employed to modify the logits, resulting in adjusted logits $\hat{o} = g(o)$.

The primary objective is to train a model that minimizes a specific classification error metric. The class-conditional errors are calculated over the data distribution as $\text{Err}_k = \mathbb{P}\left[y \neq \hat{y}_f(x) \mid y = k\right]$. The standard misclassification error is denoted by $\text{Err}_{\text{plain}} = \mathbb{P}\left[y \neq \hat{y}_f(x)\right]$. In situations with label imbalance, $\text{Err}_{\text{plain}}$ might be dominated by the majority classes. To this end, balanced classification error $\text{Err}_{\text{bal}} = (1/K) \sum_{k=1}^K \text{Err}_k$ is widely employed as a fairness metric. We will later introduce various objectives that aim to achieve different fairness goals. A comprehensive list of the objectives examined in this study can be found in Appendix A.

## 3 Proposed Approach: Class-attribute Priors (CAP)

### 3.1 Class Attributes and Adaptation to Heterogeneity

We start by introducing the CAP approach at a conceptual level and provide concrete applications of CAP to loss function design in Section 3.2. Recall that our high-level goal is designing a map from

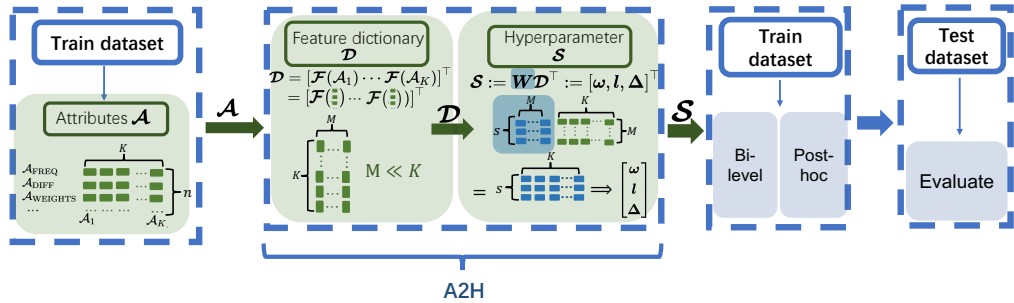

Figure 2: The overview of CAP approach. CAP is the overall framework proposed in our paper, with **A2H** being the core algorithm. **A2H** is a meta-strategy that transforms the class-attribute prior knowledge into hyper-parameter $\mathcal{S}$ for each class through a trainable matrix **W**, forming a training strategy that satisfies the desired fairness objective. The left half of the figure specifically illustrates how our algorithm calculates and trains the weights. In the first stage, we collect class-related information and construct an attribute table of $n \times K$ dimension. This is a general prior, which is related to the distribution of training data, the training difficulty of each class, and other factors. Then, he first step of **A2H** is to compute a $K \times M$ Feature Dictionary $\mathcal{D} = \mathcal{F}(\mathcal{A})$ by applying a set of functions $\mathcal{F}$. We remark that $M << K$ and $M$ is only related to the number of attributes $n$ and $|\mathcal{F}|$, making it a constant. Therefore, the search space is $\mathcal{O}(1)$. Then, in the second step, the weight matrix **W** is trained through bi-level or post-hoc methods to construct the hyperparameter $\mathcal{S}$.

**A2H** that takes attributes $\mathcal{A}_k$ of class $k$ and generates the hyperparameters of the optimization strategy $\mathcal{S}_k$. Each coordinate $\mathcal{A}_k[i]$ characterizes a specific attribute of class $k$ such as label frequency, label noise ratio, training difficulty shown in Table 1. To model **A2H**, one can use any hypothesis space including deep nets. However, since **A2H** will be optimized over the validation loss, depending on the application scenario, it is often preferable to use a simpler linearized model. **Linearized approach**. Suppose each class has $n$ attributes with $\mathcal{A}_k \in \mathbb{R}^n$. We will use a nonlinear feature map $\mathcal{F}(\cdot) : \mathbb{R}^n \to \mathbb{R}^M$ where $M$ is the embedding space. Suppose the class-specific strategy $\mathcal{S}_k \in \mathbb{R}^s$. Then, **A2H** can be parameterized by a weight matrix $\boldsymbol{W} \in \mathbb{R}^{s \times M}$ so that

$$\mathcal{S}_k = \textbf{A2H}(\mathcal{A}_k) := \boldsymbol{W}\mathcal{F}(\mathcal{A}_k). \tag{1}$$

Our goal becomes finding $\boldsymbol{W}$ so that the resulting strategies maximize the target validation objective. Observe that $\boldsymbol{W}$ has $s \times M$ parameters rather than $s \times K$ parameters which is the naive approach that learns individual strategies. In practice, $K$ can be significantly large, so for typical problems, $M \ll K$. Moreover, $\boldsymbol{W}$ ties all classes together during training through weight-sharing whereas the naive approach would be brittle for tail classes that contain very limited data. The approach are summarized in Fig. 2

### 3.2 CAP for Loss Function Design

Consider the generalized cross-entropy loss

$$\ell(y, f(\boldsymbol{x})) = \omega_y \log(1 + \sum_{k \neq y} e^{l_k - l_y} \cdot e^{\Delta_k f_k(\boldsymbol{x}) - \Delta_y f_y(\boldsymbol{x})}).$$

Here, $(\omega_k, \boldsymbol{l}_k, \Delta_k)_{k=1}^K$ are hyperparameters that can be tuned to optimize the desired test objective. For class $k$, we get to choose the tuple $\mathcal{S}_k := [\omega_k, \boldsymbol{l}_k, \Delta_k]$ which can be considered as its training strategy. Here elements of $\mathcal{S}_k$ arise from existing imbalance-aware strategies, namely weighting $\omega_k$, additive logit-adjustment $\boldsymbol{l}_k$ and multiplicative adjustment $\Delta_k$.

**Example: LA and CDT losses viewed as CAP**. For label imbalanced problems, [25, 38] propose to set hyperparameters $\boldsymbol{l}_k$ and $\Delta_k$ as a function of frequency $\pi_k = \mathbb{P}(y = k)$. Concretely, they propose

| Attributes | Definition | Notation | Application scenario |
|---|---|---|---|
| $\mathcal{A}_{\text{FREQ}}$ | Class frequency | $\pi_k = \mathbb{P}(y = k)$ | Imbalanced classes |
| $\mathcal{A}_{\text{DIFF}}$ | Class-conditional error | $\mathbb{P}(y \neq \hat{y})$ | Difficult vs easy classes |
| $\mathcal{A}_{\text{WEIGHTS}}$ | Test-time class weights | $\omega_k^{\text{test}}$ of (3) | Weighted test accuracy |
| $\mathcal{A}_{\text{NOISE}}$ | Label noise ratio | $\mathbb{P}(y^{\text{CLEAN}} \neq y \| y^{\text{clean}} = k)$ | Datasets with label noise |
| $\mathcal{A}_{\text{NORM}}$ | Norm of classifier weights | See [3] | Imbalanced classes |

Table 1: Definition of example attributes and associated application scenarios. Attributes $\mathcal{A}_{\text{DIFF}}$ and $\mathcal{A}_{\text{NORM}}$ are computed during the training (for post-hoc optimization, it is pre-training). For bilevel training they are computed at the end of warm-up. The upper attributes in red color are those we utilize in our experiments. Also we use $\mathcal{A}_{\text{ALL}}$ to denote combined attributes.

$l_k = -\gamma \log(\pi_k)$ [25] and $\Delta_k = \pi_k^\gamma$ [38] for some scalar $\gamma$. These can be viewed as special instances of CAP where we have a single attribute $\mathcal{A}_k = \pi_k$ and **A2H**$(x)$ is $-\gamma \log(x)$ or $x^\gamma$ respectively. **Our approach** can be viewed as an extension of these to attributes beyond frequency and general class of **A2H**. In light of (1), hyperparameters of a specific element of $\mathcal{S}_k = [\omega_k, l_k, \Delta_k]$ correspond to a particular row of $W \in \mathbb{R}^{3 \times M}$ since $W = [w_\omega, w_l, w_\Delta]^\top$. Our goal is then tuning the $W$ matrix over validation data. In practical implementation, we define a feature dictionary

$$\mathcal{D} = \begin{bmatrix} \mathcal{F}(\mathcal{A}_1) & \cdots & \mathcal{F}(\mathcal{A}_K) \end{bmatrix}^\top \in \mathbb{R}^{K \times M}. \tag{2}$$

Each row of this dictionary is the features associated to the attributes of class $k$. We generate the strategy vectors $\Delta, l, \omega \in \mathbb{R}^K$ (for all classes) via $\omega = \mathcal{D} w_\omega$, $\Delta = \text{sigmoid}(\sqrt{K} \frac{\mathcal{D} w_\Delta}{\|\mathcal{D} w_\Delta\|})$, $l = \mathcal{D} w_l$.

For both loss function design and posthoc optimization, we use a decomposable feature map $\mathcal{F}$. Concretely, suppose we have basis functions $(\mathcal{F}_i)_{i=1}^m$. These functions are chosen to be poly-logarithms or polynomials inspired by [25, 38]. For $i$th attribute $\mathcal{A}_k[i] \in \mathbb{R}$, we generate $\mathcal{F}(\mathcal{A}_k[i]) \in \mathbb{R}^m$ obtained by applying $(\mathcal{F}_i)_{i=1}^m$. We then stitch them together to obtain the overall feature vector $\mathcal{F}(\mathcal{A}_k) = [\mathcal{F}(\mathcal{A}_k[1])^\top \cdots \mathcal{F}(\mathcal{A}_k[m])^\top] \in \mathbb{R}^{M:=m \times n}$. We emphasize that prior approaches are special instances where we choose a single basis function and single attribute $\pi_k$.

**Which attributes to use and why multiple attributes help?** Attributes should be chosen to reflect the heterogeneity across individual classes. These include class frequency, how difficult it is to predict that class, noisy level and more. We list such potential attributes $\mathcal{A}$ in Table 1. The frequency $\mathcal{A}_{\text{FREQ}}$ is widely used to mitigate label imbalance, and $\mathcal{A}_{\text{NORM}}$ is inspired by the imbalanced learning literature [3]. However, these may not fully summarize the heterogenous nature of the problem.

For example, some classes are more difficult to learn (e.g. due to noise or inherent predictability) and require more upweighting despite containing sufficient training examples. This can be addressed by introducing $\mathcal{A}_{\text{DIFF}}$, which characterizes the predictability of classes. In Appendix F, we provide theoretical justification for how joint use of $\mathcal{A}_{\text{DIFF}}$ and $\mathcal{A}_{\text{FREQ}}$ is needed for a Gaussian Mixture model. Moreover, rather than balanced accuracy, we may wish to optimize general test objectives including weighted accuracy with variable class importance. We can declare these test-time weights as an attribute $\mathcal{A}_{\text{WEIGHTS}}$. In Appendix E, we provide theoretical justification for incorporating $\mathcal{A}_{\text{WEIGHTS}}$ by showing CAP can accomplish Bayes optimal logit adjustment for weighted error. More broadly, any class-specific meta-feature can be declared as an attribute within CAP.

**Reduced search space and increased stability**. Searching $l$ and $\Delta$ on $\mathbb{R}^K$ with very few validation samples raises the problem of unstable optimization. [23] indicates the bilevel optimization is brittle and hard to optimize. They introduce a long warm-up phase and aggregate classes with similar frequency into $g$ groups, reducing the search space to $k/g$ dimensions. However, to achieve a fine-grained loss function, $g$ cannot be very large, so the search space remains large. In our method,

with a good design of $\mathcal{D}$ (normally $n \approx 2$ and $m \approx 3$), we can utilize a constant $2mn \ll K$ that efficiently reduces the search space and provides better convergence and stability.

We remark that dictionary is a general and efficient design that can recover multiple existing successful imbalanced loss function design algorithms. For example, [25] and [38] both utilize the frequency as $\mathcal{A}$ and apply logarithm and polynomial functions as $\mathcal{F}$ on frequency to determine $l$ and $\Delta$ respectively. Moreover, let $\mathcal{A} = I_k$ and $\mathcal{F}$ be an identity function, then training $w_l, w_\Delta$ is equivalent to train $l, \Delta$ which recovers the algorithm of [23]. Despite the ability to generalize, the dictionary is more flexible and powerful since the attributes can be chosen based on the scenarios. For example, naturally, class frequency is a critical criterion in an imbalanced dataset, but classification error in early training can also be a good criterion for evaluating class training difficulty. Furthermore, some specific attributes can be introduced to noisy or partial-labeled datasets to help design a better loss function. Our empirical study elucidates the benefit of combining multiple attributes and the dictionary performance on the noisy imbalanced dataset.

### 3.3 Class-specific Learning Strategies: Bilevel Optimization and Post-hoc optimization

To instantiate CAP as a meta-strategy, we focus on two important class-specific optimization problems: loss function design via bilevel optimization and post-hoc logit adjustment. We describe them in this section and demonstrate that both methods outperform the state-of-the-art approaches. Fig. 4 illustrates how CAP is implemented under bi-level optimization and post-hoc optimization in detail.

• **Strategy 1: Loss function design via bilevel optimization**. Inspired by [23] and following our exposition in Section 3.1, we formalize the meta-strategy optimization problem as

$$\min_{w_l, w_\Delta} \mathcal{L}_{\text{val}}(w_l, w_\Delta, f) \quad \text{s.t.} \quad \min_{f} \mathcal{L}_{\text{train}}(w_l, w_\Delta, f)$$

where $f$ is the model and $\mathcal{L}_{\text{val}}, \mathcal{L}_{\text{train}}$ are validation and training losses respectively. Our goal is finding CAP parameters $w_l, w_\Delta$ that minimize the validation loss which is the target fairness objective. Following the implementation of [23], we split the training data to 80% training and 20% validation to optimize $\mathcal{L}_{\text{train}}$ and $\mathcal{L}_{\text{val}}$. The optimization process is split to two phases: the search phase that finds CAP parameters $w_l, w_\Delta$ and the retraining phase that uses the outcome of search and entire training data to retrain the model. We note that, during initial search phase, [23] employs a long *warm-up* phase where they only train $f$ while fixing $w_l, w_\Delta$ to achieve better stability. In contrast, we find that CAP either needs very short warm-up or no warm-up at all pointing to its inherent stability (due to small hyperparameter search space, as discussed in the Appendix C).

• **Strategy 2: Post-hoc optimization**. In [25, 9, 11], the author displays that the post-hoc logit adjustment can efficiently address the bias when training with imbalanced datasets. Formally, given a model $f$, a post-hoc function $g : \mathbb{R}^K \to \mathbb{R}^K$ adjusts the output of $f$ to minimize the fairness objective. Thus the final model of post-hoc optimization is $g \circ f(x)$.

**Transferability from post-hoc optimization to loss function design**. In parametric cross entropy loss $\ell(y, f(x)) = \log\left(1 + \sum_{k \neq y} e^{\Delta_k f_k(x) + l_k - \Delta_y f_y(x) - l_y}\right)$, the output logits are adjusted by $\Delta f(x) - l$ which paves the path of searching a post-hoc **A2H′** and transfer to CAP **A2H**. [25] provides the post-hoc optimization $l'$ by flipping the sign of $l$ in loss adjustment. In our approach, we search a post-hoc **A2H′** with very marginal computation cost to obtain post-hoc $l'$ and $\Delta'$, the training loss function can be transferred from post-hoc as $\ell(y, f(x)) = \log\left(1 + \sum_{k \neq y} e^{\Delta'^{-1}_k f_k(x) - l'_k - \Delta'^{-1}_y f_y(x) + l'_y}\right)$.

## 4 Experiments and Main Results

In this section, we present our experiments in the following way. Firstly, we demonstrate the performance of CAP on both loss function design via bilevel optimization and post-hoc logit adjustment

| Method | CIFAR10-LT | CIFAR100-LT | ImageNet-LT |
|---|---|---|---|
| Cross entropy | 30.45(±0.49) | 61.94(±0.28) | 55.59(±0.26) |
| Logit adjustment (LA)[25] | 21.29†(±0.43) | 58.21†(±0.31) | 52.46♯ |
| CDT[38] | 21.57†(±0.50) | 58.38†(±0.33) | 53.47♯ |
| Plain$_{\text{Bilevel}}$ (AutoBalance[23]) | 21.15♯ | 56.70♯ | 50.91♯ |
| CAP$_{\text{Bilevel}}$ : $\mathcal{A}_{\text{ALL}}$ | **20.22**(±0.35) | **56.38**(±0.19) | **49.31**(±0.34) |
| CAP$_{\text{Post-hoc}}$ : $\mathcal{A}_{\text{ALL}}$ | 20.87(±0.38) | 57.63(±0.26) | 51.46(±0.20) |

Table 2: Balanced error on long-tailed data using loss function designed via bilevel optimization. ♯: best reported results taken from [23].†: Reproduced results.

in Sec. 4.1. We further highlight the connection between bilevel CAP and post-hoc optimization by transferring the learned hyper-parameters. Sec. 4.2 demonstrates that CAP provides noticeable improvements for fairness objectives beyond balanced accuracy. Then Sec. 4.3 discusses the advantage of utilizing attributes and how CAP leverages them in noisy, long-tailed datasets through perturbation experiments. Lastly, we defer the experiment details including hyper-parameters, number of trails, and other reproducibility information to appendix.

**Dataset**. In line with previous research [25, 38], we conduct the experiments on CIFAR-LT and ImageNet-LT datasets. The CIFAR-LT modifies the original CIFAR10 or CIFAR100 by reducing the number of samples in tail classes. The imbalance factor, represented as $\rho = N_{max}/N_{min}$, is determined by the number of samples in the largest ($N_{max}$) and smallest ($N_{min}$) classes. To create a dataset with the imbalance factor, the sample size decreases exponentially from the first to the last class. We use $\rho = 100$ in all experiments, consistent with previous literature. The ImageNet-LT, a long-tail version of ImageNet used in various fairness research [25, 23], has 1000 classes with an imbalanced ratio of $\rho = 256$. The maximum and minimum samples per class are 1280 and 5, respectively. During the search phase for bilevel CAP and post-hoc transferability experiments (Sec. 4.1), we split the training set into 80% training and 20% validation to obtain the optimal loss function design. We remark that the validation set is imbalanced, with tail classes containing very few samples, making it challenging to find optimal hyper-parameters without overfitting. For all other post-hoc experiments (Sec. 4.2 and 4.3), we follow the setup of [25, 11] by training a model on entire training dataset as the pre-train model, and optimizing a logit adjustment $g$ on a balanced validation dataset. Additionally, all CIFAR-LT experiments use ResNet-32 [13], and ImageNet-LT experiments use ResNet-50, in accordance with literature.

## 4.1 CAP Improves Prior Methods Using Post-hoc or Bilevel Optimization

This section presents our loss function design experiments on imbalanced datasets by incorporating CAP into the training scheme of [23, 25, 38], as discussed in Sec. 3.3. Table 2 demonstrates our results. The first part displays the outcomes of various existing methods with their optimal hyper-parameters. It is worth noting that the original best results for single-level methods ([25, 38]) are obtained from grid search on the test dataset, which leads to much better performance than our reproduced results using validation grid search in Table. 2. Moreover, both of the grid search methods demand substantial computation budgets. As illustrated in the second part of Table 2, bilevel and post-hoc CAP significantly improve the balanced error across all datasets. We also conduct experiments to further bridge the connection between post-hoc and bilevel loss function design, as discussed in Sec.3.1, which can be found in Appendix D.

## 4.2 Benefits of CAP for Optimizing Distinct Fairness Objectives

Recent works on label-imbalance places a significant emphasis on the balanced accuracy evaluations [25, 23, 3]. However, in practice, there are many different fairness criteria and balanced accuracy is only one of them. In fact, as we discuss in (3), we might even want to optimize arbitrary weighted test objectives. In this section, we demonstrate the flexibility and merits of CAP when optimizing

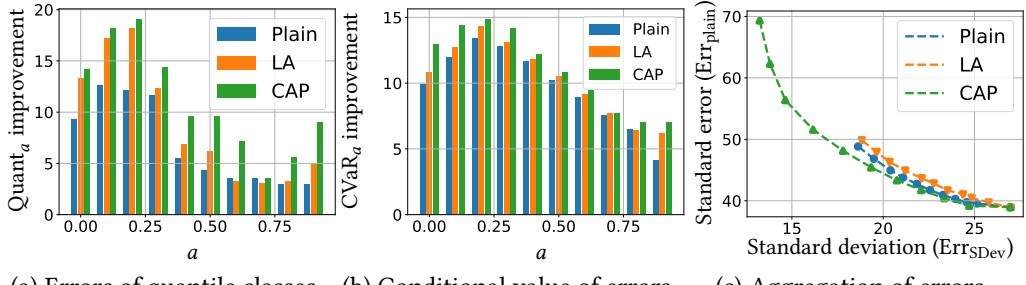

(a) Errors of quantile classes    (b) Conditional value of errors    (c) Aggregation of errors

Figure 3: Benefit of CAP for optimizing different Fairness Objectives. We compare among plain post-hoc, LA post-hoc and $\text{CAP}_{\text{post-hoc}}$. (a): Results of optimizing quantile class performance $\text{Quant}_a = \mathbb{P}\left[y \neq \hat{y}_f(\boldsymbol{x}) \mid y = \text{K}_a\right]$, where $\text{K}_a$ denotes the class index with the worst $\lceil \text{K} \times a \rceil$-th error. (b): Results of optimizing tail performance $\text{CVaR}_a$. (c): Results of optimizing $\mathcal{R}(\text{Err}) = \lambda \cdot \text{Err}_{\text{plain}} + (1 - \lambda) \cdot \text{Err}_{\text{SDev}}$. The plot shows the trade-off between standard deviation of class-conditional errors $\text{Err}_{\text{SDev}}$ and Standard misclassification error $\text{Err}_{\text{plain}}$ as $\lambda$ varies. See Sec.4.2 for detailed definition and discussions.

| Post-hoc methods | $\text{Err}_{\text{bal}}$ | $\text{Err}_{\text{SDev}}$ | $\text{CVaR}_{0.2}$ | $\text{Quant}_{0.2}$ | $\text{Err}_{\text{weighted}}$ |
|---|---|---|---|---|---|
| Pretrained | 61.94 ($\pm$0.28) | 27.13($\pm$0.35) | 96.95($\pm$0.15) | 93.01($\pm$0.58) | 62.53($\pm$0.53) |
| $\text{Plain}_{\text{Post-hoc}}$ | -1.62($\pm$0.36) | -8.51($\pm$0.75) | -11.48($\pm$0.81) | -12.79($\pm$0.43) | -2.82($\pm$0.56) |
| $\text{LA}_{\text{Post-hoc}}$ | -3.73($\pm$0.29) | -8.72($\pm$0.66) | -12.21($\pm$0.50) | -15.01($\pm$0.35) | -3.62($\pm$0.37) |
| $\text{CAP}_{\text{Post-hoc}}$ | **-4.36($\pm$0.25)** | **-13.92($\pm$0.24)** | **-14.75($\pm$0.87)** | **-18.34($\pm$0.47)** | **-6.21($\pm$0.49)** |

Table 3: The error difference between other approaches compared to pre-trained model. The first line shows the performance of Pretrained model, and the following line shows the error difference of other methods (smaller is better). For objectives with $a$, we set $a = 0.2$. This is commonly used for difficult or few classes in other papers[40, 24].

fairness objectives other than balanced accuracy. The experiments are conducted on the CIFAR100-LT dataset using the post-hoc approaches. For the fairness objectives, we mainly focus on three objectives: quantile class error $\text{Quant}_a$, conditional value at risk (CVaR) $\text{CVaR}_a$, and the combined risk $\mathcal{R}(\text{Err})$, which consists of standard deviation of error and the regular classification error.

To begin with, we first demonstrate the performance on quantile class error $\text{Quant}_a = \mathbb{P}\left[y \neq \hat{y}_f(\boldsymbol{x}) \mid y = \text{K}_a\right]$, where $\text{K}_a$ denotes the class index with the worst $\lceil \text{K} \times a \rceil$-th error. For instance, in CIFAR100-LT, where $K = 100$, $\text{Quant}_{0.2}$ denotes the test error of the worst 20 percentile class. That is, we sort the classes in descending order of test error and return the error of the class 20%th class ID. Thus, each selection of $a$ raises a new objective. Fig. 3a shows the improvement over the pre-trained model when optimizing $\text{Quant}_a$ with multiple selections of $a$. We observe that CAP significantly outperforms both logit adjustment and plain post-hoc.

Moreover, the $\text{CVaR}_a = \mathbb{E}\left[\text{Err}_k \mid \text{Err}_k > \text{Quant}_a\right]$ measures the average error of $\lceil \text{K} \times a \rceil$ classes with worst errors. Instead of $\text{Quant}_a$, which only focuses on the specific quantile class error, optimizing the $\text{CVaR}_a$ tend to improve the tail behavior of the classifier, which is a more general fairness objective. Fig. 3b shows the test improvements over three approaches, and CAP is consistently better than all other methods.

Finally, for the combined risk $\mathcal{R}(\text{Err})$, we define $\mathcal{R}(\text{Err}) = \lambda \cdot \text{Err}_{\text{plain}} + (1 - \lambda) \cdot \text{Err}_{\text{SDev}}$ where $\text{Err}_{\text{plain}}$ is the regular classification error and $\text{Err}_{\text{SDev}}$ denotes the standard deviation of classification errors. We plot the error-deviation curve by varying $\lambda$ from 0 to 1 with stepsize 0.1 on three approaches in Fig. 3c, each point corresponds to a different $\lambda$. We observe that plain post-hoc

| | CIFAR100-LT | | ImageNet-LT | | CIFAR10-LT+Noise | |
|---|---|---|---|---|---|---|
| | $\text{Err}_{\text{bal}}$ | $\text{Err}_{\text{SDev}}$ | $\text{Err}_{\text{bal}}$ | $\text{Err}_{\text{SDev}}$ | $\text{Err}_{\text{bal}}$ | $\text{Err}_{\text{SDev}}$ |
| Cross entropy | 61.94(±0.28) | 27.13(±0.35) | 55.59(±0.26) | 29.10(±0.64) | 43.76(±0.74) | 31.69(±0.81) |
| $\text{Plain}_{\text{Bilevel}}$ (AutoBalance [23]) | 56.70(±0.32) | 20.13(±0.68) | 50.93(±0.16) | 26.06(±0.61) | 40.04(±0.79) | 36.30(±0.89) |
| $\text{CAP}_{\text{Bilevel}}$: $\mathcal{A}_{\text{FREQ}}$ | 56.64 (±0.21) | 19.10(±0.67) | 50.82(±0.13) | 24.36(±0.49) | 39.91(±0.66) | 26.54(±0.80) |
| $\text{CAP}_{\text{Bilevel}}$: $\mathcal{A}_{\text{DIFF}}$ | 58.27(±0.24) | **17.62(±0.65)** | 52.97(±0.30) | **21.28(±0.58)** | 40.61(±0.61) | **14.49(±0.72)** |
| $\text{CAP}_{\text{Bilevel}}$: $\mathcal{A}_{\text{FREQ}}$+$\mathcal{A}_{\text{DIFF}}$ | **56.38(±0.19)** | 18.53(±0.63) | **49.31(±0.34)** | 22.14(±0.46) | **38.36(±0.79)** | 19.78(±0.75) |

Table 4: Attributes help optimization adapt to dataset heterogeneity. We conduct experiments using bilevel loss design and report the balanced misclassification error,and standard deviation of class-conditional errors with different class-specific attributes.

cannot achieve a small standard deviation, and post-hoc LA degrades when achieving smaller $\text{Err}_{\text{SDev}}$, CAP accomplish the best performance and are flexible to adapt to different objectives.

Regarding the plain post-hoc in Fig. 3, we find that without class-specific attribute prior information, the parameter of each class is updated individually. Optimizing towards a specific objective (e.g., $\text{Quant}_a$) may dramatically hurt the performance of other classes and cause the changing of under-represented classes. Thus, the plain post-hoc optimization is unstable, and hard to achieve good results. On the other hand, although post-hoc LA outperforms plain post-hoc, optimizing only one temperature variable lacks fine-grained adaptation to various objectives. In contrast, CAP exhibits a noticeably better performance on all objectives since CAP takes both class-specific attribute and fine-grained control into consideration.

Table 3 shows more results. $\text{Err}_{\text{weighted}}$ denotes a weighted test objective induced by weights $\omega_k^{\text{test}} \in \mathbb{R}^K$ given by

$$\text{Err}_{\text{weighted}} = \sum_{k=1}^{K} \omega_k^{\text{test}} \text{Err}_k \quad \text{where} \quad \sum_{k=1}^{K} \omega_k^{\text{test}} = K. \tag{3}$$

Overall, Table 3 shows that CAP consistently achieves the best results on multiple fairness objectives. An important conclusion is that, the benefit of CAP is more significant for objectives beyond balanced accuracy and improvements are around 2% or more (compared to [25] or plain post-hoc). This is perhaps natural given that prior works put an outsized emphasis on balanced accuracy in their algorithm design [25, 23].

### 4.3 Benefits of CAP for Adapting to Distinct Class Heterogeneities

Continuing the discussion in Sec. 3.1, we investigate the advantage of different attributes in the context of dataset heterogeneity adoption. In Table 4, we conduct loss function design CAP experiments on CIFAR-LT and ImageNet-LT dataset. Specifically, besides using regular CIFAR100-LT and ImageNet-LT, we introduce label noise into CIFAR10-LT following [34, 29] to extend the heterogeneity of the dataset. To add the label noise, firstly, we split the training dataset to 80% train and 20% validation to accommodate bilevel optimization. Then we randomly generate a noise ratio $r \in \mathbb{R}^K, r_i \sim U(0, 0.5)$ that denotes the label noise ratio for each class. Finally, keeping the validation set clean, we add label noise into the train set by randomly flipping the labels of selected training samples (according to the noise ratio) to all possible labels, which is the same as literature[34, 29]. As a result, all classes contain an unknown fraction of label noise in the noisy CIFAR10-LT dataset, which raises more heterogeneity and challenge in optimization. Through bilevel optimization, we optimize the balanced classification loss and report the balanced test error and its standard deviation after the retraining phase in Table 4. As shown in Table 4, we employ label frequency $\mathcal{A}_{\text{FREQ}}$ which is designed for sample size heterogeneity and $\mathcal{A}_{\text{DIFF}}$ which is designed for class predictability as the attributes in CAP approach. Table 4 highlights that CAP consistently outperforms other methods while different attributes can shape the optimization process differently. Importantly, CAP is particularly favorable to tail classes which contain too few examples to optimize individually. Only using $\mathcal{A}_{\text{DIFF}}$ achieves smallest $\text{Err}_{\text{SDev}}$ demonstrating that optimization with $\mathcal{A}_{\text{DIFF}}$

tends to keep better class-wise fairness because $\mathcal{A}_{\mathrm{DIFF}}$ is directly related to class predictability. The combination of $\mathcal{A}_{\mathrm{FREQ}}$ and $\mathcal{A}_{\mathrm{DIFF}}$ shows that incorporating multiple class-specific attributes provides additional information about the dataset and jointly enhances performance. Overall, the results indicate that CAP establishes a principled approach to adapt to multiple kinds of heterogeneity.

## 5 Conclusions

This paper proposed a new meta-strategy CAP to tackle class heterogeneities and general fairness objectives. CAP achieves high-validation performance by efficiently generating class-specific strategies based on various class attributes. Applications and experiments on posthoc optimization and loss function design demonstrate that CAP substantially improves multiple types of fairness objectives as well as general weighted test objectives. We also demonstrate the transferability across our strategies: Posthoc CAP can be plugged in as a loss function to further boost accuracy. **Broader impacts**. Although our approach and applications primarily focus on loss function design and posthoc optimization, CAP approach can also help design class-specific data augmentation, regularization, and optimizers. Additionally, rather than heterogeneities across classes, one can extend CAP-style personalization to problems in multi-task learning and recommendation systems. **Limitations**. Observe that, if we have infinite training data, we can search for optimal strategies for each class. Thus, the primary limitation of CAP is its multi-task design space that shares the same meta-strategy across classes. However, as experiments demonstrate, in practical finite data settings, CAP achieves better data efficiency and test performance compared to individual tuning.

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

Figure 4: CAP framework for detailed implementation. This figure illustrates how CAP is implemented under bi-level optimization and post-hoc optimization. Throughout the entire figure, the only trainable parameters are **W** and the network (in the green box). In the search phase of bilevel optimization, we first conduct an 80-20% train-val split. Then, we train the network with parametric loss function for inner optimization on 80% training dataset and train **W** to achieve fairness objective for outer optimization on 20% validation dataset. And in post-hoc implementation, we first train the network without hyperparameters on the training dataset and do the post-hoc optimization on the validation set. Both bilevel and post-hoc yield optimal fairness weight $\mathbf{W}^*$, for bi-level and post-hoc transferring, we use the optimal $\mathbf{W}^*$ to retrain a fairness-focused model on the entire training dataset. If only post-hoc adjustments are conducted, we directly modify the pre-trained model's logit with a post-hoc function.

## A List of fairness objectives

We list all the notation of objectives we used in the main paper in this section.

| Symbol | Meaning |
|---|---|
| $\ell, f$ | Loss function (specifically cross-entropy), predictor |
| $\mathrm{Err}(f)$ | Error of $f$ on entire population |
| $\mathrm{Err}_k$ | Class-conditional error of $f$ on class K = k |
| $\mathrm{Err}_{\mathrm{plain}}$ | Standard misclassification error |
| $\mathrm{Err}_{\mathrm{bal}}$ | Balanced misclassification error, average of class-conditional errors |
| $\mathrm{Err}_{\mathrm{weighted}}$ | Weighted misclassification error |
| $\mathrm{Err}_{\mathrm{SDev}}$ | Standard deviation of class-conditional errors |
| $\mathrm{Quant}_a$ | Errors of quantile classes at level $a$ |
| $\mathrm{CVaR}_a$ | Conditional value of errors at level $a$ |
| $\mathcal{R}(\mathrm{Err})$ | Aggregation of class-conditional errors |

## B Framework overview.

## C Extended Discussion of Warm-up and Training Stability

In Sec. 3.2, we discuss how CAP stabilizes the training and eases the necessarily of warm-up. Now, we extend the discussion and provide more experiments to demonstrate further the benefit of the CAP strategy in this section. In Table 5, we conduct experiments on bilevel loss function design on CIFAR10-LT. Firstly, we investigate the performance of the default initialization (DI) of $\mathrm{Plain}_{Bilevel}$

|  | DI, 100 epoch | DI, 120 epoch | DI, 200 epoch | LA , 120 epoch | Self-sup[5] |
|---|---|---|---|---|---|
| $\text{Err}_{\text{SDev}}$ when search phase begin | 0.23 | 0.20 | 0.28 | 0.17 | 0.13 |
| $\text{Err}_{\text{bal}}$ | 24.58 | 21.39 | 23.36 | 21.15 | 20.57 |

Table 5: Bilevel training with different warm-up lead to different result on CIFAR10-LT. We investigate the performance of the default initialization (DI) of $\text{Plain}_{bilevel}$ where $l = 0$ and $\Delta = 1$ with 100,120 and 200 warm-up epochs, and we also provide the result where $l$ starts with logit adjustment prior. We implement the self-supervision pre-trained model by SimSiam [5]. We remark that 120 epochs Warm-up with DI or LA loss are used in [23].

|  |  | CIFAR10-LT | | CIFAR100-LT | |
|---|---|---|---|---|---|
|  |  | search phase | retrain | search phase | retrain |
| Post-hoc LA[25] |  | 21.43($\pm$0.30) | 22.34($\pm$0.34) | 58.48($\pm$0.23) | 57.65($\pm$0.25) |
| Post-hoc CDT[38] |  | 23.58($\pm$0.37) | 21.79($\pm$0.40) | 58.60($\pm$0.26) | 57.86($\pm$0.27) |
| $\text{Plain}_{\text{Post-hoc}}$ | $l$ | 20.90($\pm$0.28) | 21.71($\pm$0.29) | 57.98($\pm$0.22) | 57.82($\pm$0.19) |
|  | $\Delta$ | 23.74($\pm$0.34) | 24.06($\pm$0.36) | 58.61($\pm$0.29) | 58.80($\pm$0.31) |
|  | $l\&\Delta$ | 23.41($\pm$0.30) | 23.38($\pm$0.33) | 57.80($\pm$0.24) | 58.57($\pm$0.23) |
| $\text{CAP}_{\text{Post-hoc}}$ | $w_l$ | 20.81 ($\pm$0.15) | 20.65($\pm$0.36) | 57.73($\pm$0.25) | 57.15($\pm$0.30) |
|  | $w_\Delta$ | 22.31($\pm$0.38) | 21.06($\pm$0.43) | 58.07($\pm$0.32) | 57.26($\pm$0.35) |
|  | $w_l \& w_\Delta$ | 20.87($\pm$0.38) | 20.32($\pm$0.64) | 57.63($\pm$0.26) | 57.08($\pm$0.21) |

Table 6: Balanced error on long-tailed data using post-hoc logits adjustment. The search phase results reveal the test accuracy of post-hoc adjustment, which is searched on a 20% validation set. The retrain results show the transferability from post-hoc logits adjustment to loss function design.

where $l = 0$ and $\Delta = 1$ with 100,120 and 200 warm-up epochs. Then we provide the result where $l$ starts with logit adjustment prior. Finally, we implement the self-supervision pre-trained model by SimSiam [5]. Table 5 presents the relationship between the $\text{Err}_{\text{SDev}}$ of the pre-trained model and the final $\text{Err}_{\text{bal}}$ after bilevel training. One direct observation is that $\text{Err}_{\text{SDev}}$ highly correlates with $\text{Err}_{\text{bal}}$. Considering $\text{Err}_{\text{SDev}}$ measures the fairness of the pre-trained model, we believe that a better pre-trained model promotes the test performance accordingly.

Moreover, regarding LA initialization, one can conclude that initializing the training with a designed loss such as LA loss can significantly improve the result. Still, it requires additional effort and expertise in designing that specific loss, especially when the fairness objective is not only balanced error and various heterogeneities exist in the data. While the self-supervised pre-trained model achieves the best $\text{Err}_{\text{SDev}}$ and $\text{Err}_{\text{bal}}$ among all methods, training the self-supervision model requires a long time. Our proposed $\text{CAP}_{Bilevel}$, which utilizes the attributes, not only ensures to take advantage of prior knowledge but also stabilizes the optimization by simultaneously updating weights of all classes thanks to the dictionary design. $\text{CAP}_{Bilevel}$ achieves **20.16** $\text{Err}_{\text{bal}}$ on CIFAR10-LT and **56.55** $\text{Err}_{\text{bal}}$ on CIFAR100-LT with only **5** epochs of warm-up, which improves on both computation efficiency and test performance.

# D  Further post-hoc discussion

**Connection to post-hoc adjustment** To better understand the potential of CAP and the connection between loss function design and post-hoc adjustment, we design an experiment with results shown in Table 6. In this experiment, we use the same dictionary , split the original training data to 80% train and 20% validation, and train a model $f$ using regular cross-entropy loss on the 80% train set as the pre-trained model, which is biased toward the imbalanced distribution. Our goal is to

find a post-hoc adjustment $g$ so that $g \circ f$ achieves minimum balanced loss on the 20% validation set. In Table 6, the searching phase displays the test error of adjusted model $g \circ f$. Following the transferability discussion in Sec. 3.3, we use the searched post-hoc adjustment as the loss function design to retrain the model from scratch on the entire training dataset. Interestingly, retraining further improves the post-hoc performance. As post-hoc adjustment requires only about 1/5 of the time and fewer computational resources than loss function design, it provides a simple and efficient approach for loss function design.

We also observe that training $w_l$ along with $w_\Delta$ leads to performance degradation compared to only training $w_l$, and training only $w_\Delta$ also performs worse than $w_l$. We conduct more experiments and provide explanations for this. In each part of the Table 6, we compare the performance of optimizing $l$ and $\Delta$ in the similar setup, for example, LA provides a design of $l$ while CDT adjusts the loss by design a specific $\Delta$. Among all the methods, optimizing $l$ or $w_l$ always achieve the best result. We observe a degeneration when optimizing only $\Delta$ or both $l\&\Delta$. Through this section, Fig. 5 and 6 exhibit some insights and intuitions towards this phenomenon.

Fig. 5 shows the logits value before and after post-hoc adjustment. Without proper early-stopping or regularization, $\Delta$ in Fig. 5c will keep increasing and result in a stretched logits distribution, where the logits become larger and larger. Note that Fig 5c stops after 500 epochs, but longer training will even further enlarge the logits. Furthermore, because the data is not linear separable, $\Delta$ may reduce the loss in unexpected ways. The mismatch between test loss and balanced test error in Fig. 6b verified this conjecture. The loss decreases at the end of the training while the balanced error increases. That might happen because $\Delta$ performs a multiplicative update on logits as shown in Fig. 5c. Finally, the logits value becomes much larger, but the improvement is limited. Lemma 1 in paper [23] also offers possible explanation by proving loss function is not consistent for standard or balanced errors if there are distinct multiplicative adjustments i.e. $\Delta_i \neq \Delta_j$ for some $i, j \in [K]$.

In sum, the main difference between using the two different hyperparameters for post-hoc logit adjustment is that $l$ performs an additive update on logits, however, $\Delta$ performs a multiplicative update. That will leads to different behaviors. For example, if there is a true but rare label $k = i$ with negative logits value $o_i$; meanwhile, there are other labels with positive or negative values, multiplicative update using $\Delta$ couldn't help label $k$ changes the class because the logits is already negative. For post-hoc logit adjustment using $l$, it can eliminate the influence of the original value. Smaller values of $l_i$ could always make $\hat{o}_i = o_i - l_i$ have a larger boost than $\hat{o}_{k \neq i}$ Fig.5 indeed shows that there exist many samples like this.

# E  Proofs of Fisher Consistency on Weighted Loss

For more insight of the weighted test loss we discussed in Sec. 4.3, [25] proposes a family of Fisher consistent pairwise margin loss as

$$\ell(y, f(x)) = \alpha_y \cdot \log[1 + \sum_{y' \neq y} e^{\Delta_{yy'}} \cdot e^{f_{y'}(x) - f_y(x)}]$$

where pairwise label margins $\Delta_{yy'}$ denotes the desired gap between scores for $y$ and $y'$. Logit adjustment loss [25] corresponds to the situation where $\alpha_y = 1$ and $\Delta_{yy'} = \log \frac{\pi_{y'}}{\pi_y}$ where $\pi_y = \mathbb{P}(y)$. They establish the theory showing that there exists a family of pairwise loss, which Fisher consistent with balanced loss when $\Delta_{yy'} = \log \frac{\alpha_{y'} \pi_{y'}}{\alpha_y \pi_y}$ for any $\alpha \in \mathbb{R}_+^K$. However, Sec. 4.3 focuses on the weighted loss which is more general and formulated as following.

$$\ell_\omega(y, f(x)) = \alpha_y \cdot \omega_y^{test} \log[1 + \sum_{y' \neq y} e^{\Delta_{yy'}} \cdot e^{f_{y'}(x) - f_y(x)}] \tag{4}$$

Following [25], Thm. 1 deduces the family of Fisher-consistent loss with weighted pairwise loss. The followed discussion demonstrates that CAP using $\mathcal{A}_{\text{FREQ}}$ and $\mathcal{A}_{\text{WEIGHTS}}$ is able to recover Fisher-consistent loss for any $\omega^{test}$.

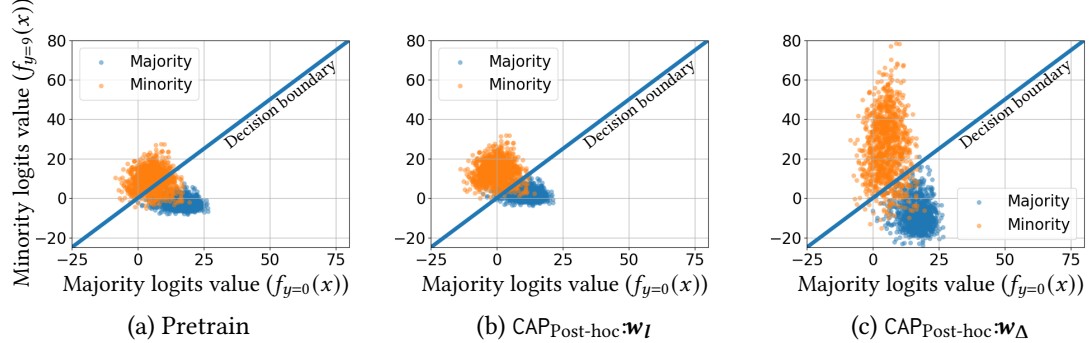

(a) Pretrain  (b) CAP$_{\text{Post-hoc}}$:$\boldsymbol{w_l}$  (c) CAP$_{\text{Post-hoc}}$:$\boldsymbol{w_\Delta}$

Figure 5: The evolution of logits in post-hoc logits adjustment CAP when optimizing $\boldsymbol{w_l}$ and $\boldsymbol{w_\Delta}$ individually. In this experiment, we train a ResNet-32 as the pre-trained model on CIFAR10-LT, where the class $y = 0$ has the largest sample size and $y = 9$ has the smallest sample size when training. In Fig. 5a, we plot the logits value $f_y^{test}(x)$ of **test dataset**. Specifically, for better visualization and understanding, we only pick two classes, the largest class ($y = 0$) as majority and the smallest class ($y = 9$) as minority. The x-axis is the logit value of majority class $f_{y=0}(x)$ and the y-axis is the logit value of minority class $f_{y=9}(x)$. Thus, the blue line ($y = x$) can be treated as the decision boundary between the two classes. In Fig. 5b shows the logits after CAP$_{\text{Post-hoc}}$ with only optimizing $\boldsymbol{w_l}$ and Fig. 5b shows the logits after CAP$_{\text{Post-hoc}}$ that only optimizing $\boldsymbol{w_\Delta}$. For clarification, the logits are directly picked from CIFAR10-LT classification problem which are not binary classification logits. We also remark that any choice of majority and minority that satisfies $N_{\text{majority}}^{\text{train}} > N_{\text{minority}}^{\text{train}}$ shows the similar result even under another training distribution differed from CIFAR10-LT (e.g. flipping the minority and majority).

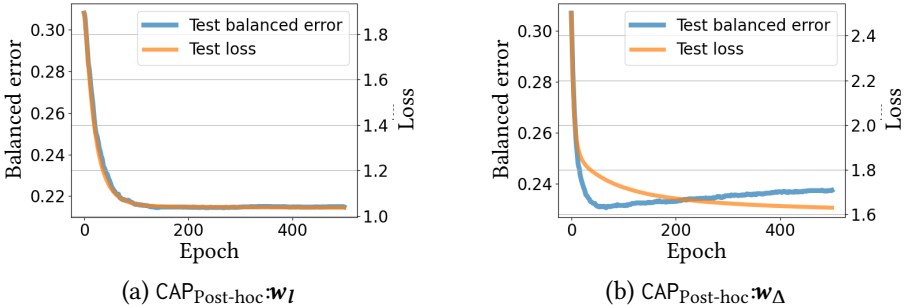

(a) CAP$_{\text{Post-hoc}}$:$\boldsymbol{w_l}$  (b) CAP$_{\text{Post-hoc}}$:$\boldsymbol{w_\Delta}$

Figure 6: Test error and loss during CIFAR10-LT post-hoc training. In Fig. 6a we only optimize $\boldsymbol{w_l}$ and we observe that balanced test error decreases with test loss simultaneously. However, in Fig. 6b where we only optimize $\boldsymbol{w_\Delta}$, the test loss (the orange curve) is keeping decreasing, but test balanced error (the blue curve) first reaches minimum and then increases. This mismatch together with Fig. 5 further explain the reason of degeneration when optimize $\boldsymbol{w_\Delta}$ by post-hoc.

**Theorem 1.** *For any $\delta \in \mathbb{R}^K_+$, the weighted pairwise loss (4) is Fisher consistent with weights and margins*

$$\alpha_y = \frac{\delta_y}{\pi_y} \quad \Delta_{yy'} = log(\delta'_y/\delta_y)$$

*Proof.* Suppose we use margin $\Delta_{yy'} = \log \frac{\delta_{y'}}{\delta_y}$, the weighted loss become

$$
\begin{aligned}
\ell_\omega(y, f(x)) &= -\omega_y^{test} \log \frac{\delta_y e^{f_y(x)}}{\sum_{y' \in [K]} \delta_{y'} e^{f_{y'}(x)}} \\
&= -\omega_y^{test} \log \frac{e^{f_y(x)+\log(\delta_y)}}{\sum_{y' \in [K]} e^{f_{y'}(x)+\log(\delta_{y'})}}
\end{aligned}
$$

Let $\mathbb{P}_\omega(y \mid x) \propto \omega_y \mathbb{P}(y \mid x)$ denote the distribution with weighting $\omega$. The Bayes-optimal score of the weighted pairwise loss will satisfy $f_y^*(x) + \log(\delta_y) = \log \mathbb{P}_\omega(y \mid x)$, which is $f_y^*(x) = \log \frac{\mathbb{P}_\omega(y|x)}{\delta_y}$.

Suppose we have a generic weights $\alpha \in \mathbb{R}^K_+$, the risk with weighted loss can be written as

$$
\begin{aligned}
\mathbb{E}_{x,y}\left[\ell_{\omega,\alpha}(y, f(x))\right] &= \sum_{y \in [L]} \pi_y \cdot \mathbb{E}_{x|y=y}\left[\alpha_y \ell_\omega(y, f(x))\right] \\
&= \sum_{y \in [L]} \pi_y \alpha_y \cdot \mathbb{E}_{x|y=y}\left[\ell_\omega(y, f(x))\right] \\
&\propto \sum_{y \in [L]} \bar{\pi}_y \cdot \mathbb{E}_{x|y=y}[\ell_\omega(y, f(x))]
\end{aligned}
$$

where $\bar{\pi}_y \propto \pi_y \alpha_y$. That means by modify the distribution base to $\bar{\pi}$, learning with the $\omega$ and $\alpha$ weighted loss 4 is equivalent to learning with the $\omega$ weighted loss. Under such a distribution, we have class-conditional distribution.

$$\overline{\mathbb{P}}(y \mid x) = \frac{\mathbb{P}_\omega(x \mid y) \cdot \bar{\pi}_y}{\overline{\mathbb{P}}(x)} = \mathbb{P}_\omega(y \mid x) \cdot \frac{\bar{\pi}_y}{\pi_y} \cdot \frac{\mathbb{P}_\omega(x)}{\overline{\mathbb{P}}(x)} \propto \mathbb{P}_\omega(y \mid x) \cdot \alpha_y \omega_y^{test}$$

Then for any $\delta \in \mathbb{R}^K_+$, let $\alpha = \frac{\delta_y}{\pi_y}$, the Bayes-optimal score will satisfy $f_y^*(x) = \log \frac{\overline{\mathbb{P}}(y|x)}{\delta_y} = \log \frac{\mathbb{P}_\omega(y|x)}{\pi_y} + C(x)$ where $C(x)$ does not depend on $y$. Thus, $\text{argmax}_{y \in [L]} f_y^*(x) = \text{argmax}_{y \in [L]} \frac{\mathbb{P}_\omega(y|x)}{\pi_y}$, which is the Bayes-optimal prediction for the weighted error.

In conclusion, there is a consistent family of weighted pairwise loss by choose any set of $\delta_y > 0$ and letting

$$\alpha_y = \frac{\delta_y}{\pi_y}$$

$$\Delta_{yy'} = \log \frac{\delta_{y'}}{\delta_y}.$$

$\square$

**Corollary 1.1.** *In* CAP, *setting attributes as* $[\mathcal{A}_{FREQ}, \mathcal{A}_{WEIGHTS}]$, $\mathcal{F} = [log(\cdot)]$. *When* $w_l = [1, -1]$, CAP *fully recovers a loss (5), which is Fisher-consistent with weighted pairwise loss.*

*Proof.* This result can be directly deduced by setting $\delta_y = \frac{\pi_y}{\omega_y^{test}}$. We have

$$\alpha_y = 1/\omega_y^{test} \quad \text{and} \quad \Delta_{yy'} = \frac{\pi_{y'}\omega_y^{test}}{\pi_y\omega_{y'}^{test}}$$

Then the corresponding logit-adjusted loss which is Fisher-consistent with weighted pairwise loss is

$$\ell(y, f(x)) = -\alpha_y\omega_y^{test} \log \frac{\delta_y \cdot e^{f_y(x)}}{\sum_{y'\in[L]} \delta_{y'} \cdot e^{f_{y'}(x)}} = -\log \frac{e^{f_y(x)+\log\pi_y-\log\omega_y^{test}}}{\sum_{y'\in[L]} e^{f_{y'}(x)+\log\pi_{y'}-\log\omega_{y'}^{test}}}. \tag{5}$$

For aforementioned CAP setup, we have $\mathcal{D}=[\log(\pi), \log(\omega_y^{test})]$, so the CAP adjusted loss with $w_l = [1, -1]$ is

$$\ell_{\text{CAP}}(y, f(x)) = -\log \frac{e^{f_y(x)+\log\pi_y-\log\omega_y^{test}}}{\sum_{y'\in[L]} e^{f_{y'}(x)+\log\pi_{y'}-\log\omega_{y'}^{test}}}. \tag{6}$$

Which is exactly the same as 5.

$\square$

# F  Multiple Attributes Benefit Accuracy in GMM

In this section, we give a simple theoretical justification why multiple attributes acting synergistically can favor accuracy. To illustrate the point, we consider a binary Gaussian mixture model(GMM), where data from the two classes are generated as follows:

$$y = \begin{cases} +1 & \text{, with prob. } \pi \\ -1 & \text{, with prob. } 1 - \pi \end{cases} \quad \text{and} \quad \mathbf{x}|y \sim \mathcal{N}(y\boldsymbol{\mu}, \sigma_y\mathbf{I}_d). \tag{7}$$

Note here that the two classes can be imbalanced depending on the value of $\pi \in (0, 1)$, which models class frequency. Also, the two classes are allowed to have different noise variances $\sigma_{\pm 1}$. This is our model for the difficulty attribute: examples generated from the class with highest variance are "more difficult" to classify as they fall further apart from their mean. Intuitively, a "good" classifier should account for both attributes. We show here that this is indeed the case for the model above.

Our setting is as follows. Let $n$ IID samples $(\mathbf{x}_i, y_i)$ from the distribution defined in (7). Without loss of generality, assume class $y = +1$ is minority, i.e. $\pi < 1/2$. We train linear classifier $(\mathbf{w}, b)$ by solving the following cost-sensitive support-vector-machines (CS-SVM) problem:

$$(\hat{\mathbf{w}}_\delta, \hat{b}_\delta) := \arg\min_{\mathbf{w},b} \|\mathbf{w}\|_2 \quad \text{sub. to } y_i(\mathbf{x}_i^T\mathbf{w} + b) \geq \begin{cases} \delta & y_i = +1 \\ 1 & y_i = -1 \end{cases}. \tag{8}$$

Here, $\delta$ is a hyperparameter that when taking values larger than one, it pushes the classifier towards the majority, thus giving larger margin to the minorities. In particular, setting $\delta = 1$ recovers the vanilla SVM. The reason why CS-SVM is particularly relevant to our setting is that it relates closely to the VS-loss. Specifically, [19] show that in linear overparameterized (aka $d > n$) settings the VS-loss with multiplicative weights $\Delta_{\pm}1$ leads to same performance as the CS-SVM with $\delta = \Delta_+/\Delta_-$. Finally, given CS-SVM solution $(\hat{\mathbf{w}}_\delta, \hat{b}_\delta)$, we measure balanced error as follows:

$$\mathcal{R}_{\text{bal}}(\delta) := \mathbb{P}_{(\mathbf{x},y)\sim(7)} \left\{ y(\mathbf{x}^T\hat{\mathbf{w}}_\delta + \hat{b}_\delta) > 0 \right\}.$$

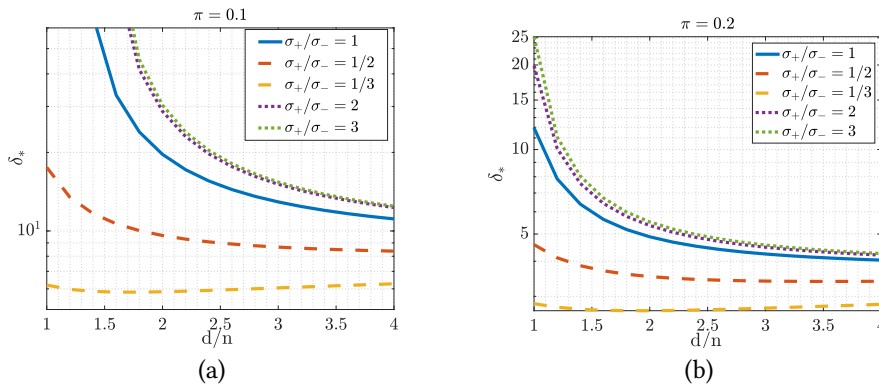

Figure 7: The optimal hyperparameter depends on both attributes: frequency ($\pi$) and difficulty ($\sigma_+/\sigma_-$).

We ask: How does the optimal CS-SVM classifier (i.e, the optimal hyperparameter $\delta$) depend on the data attributes, i.e. on the frequency $\pi$ and on the difficulty $\sigma_+1/\sigma_-1$? To answer this we consider a high-dimensional asymptotic setting in which $n, d \to \infty$ at a linear rate $d/n =: \bar{d}$. This regime is convenient as previous work has shown that the limiting behavior of the balanced error $\mathcal{R}_{\text{bal}}(\delta)$ can be captured precisely by analytic formulas [7, 27]. Specifically, [19] used that analysis to compute formulas for the optimal hyperparameter $\delta$. However, they only discussed how $\delta$ varies with the frequency attributed and only studied scenarios where both classes are equally difficult, i.e. $\sigma_{+1} = \sigma_{-1}$. Our idea is to extend their study to investigate a potential synergistic effect of frequency and difficulty.

Figure 7 confirms our intuition: the optimal hyperparameter $\delta_*$ depends both on the frequency and on the difficulty. Specifically, we see in both Figures 7(a,b) that the easier the minority class (aka, the smaller ratio $\sigma_{+1}/\sigma_{-1}$), $\delta$ decreases. That is, there is less need to favor much larger margin to the minority. On the other hand, as $\sigma_{+1}/\sigma_{-1}$ increases and minority becomes more difficult, even larger margin is favored for it. Finally, comparing Figures 7(a) to Figure 7(b), note that $\delta_*$ takes larger values for larger imbalance ratio (i.e., smaller frequency $\pi$), again aggreeing with our intuition.

## G Experiment details and reproducibility

The functions are always fixed regardless of the datasets and objectives change $\mathcal{F} = [\log(\mathcal{A}), \mathcal{A}, \mathcal{A}^\beta, \mathcal{A}^{2\beta}, \mathcal{A}^{4\beta}]$. In our experiments, we set $\beta = 0.075$.

For reproduced result in Table 2, we grid search on validation dataset and retrain for fair comparison, so the result is worse than the value reported in [25, 38] which are grid searched on whole test dataset.

For bilevel training, following the training process in [23], we start the validation optimization after 120 epochs warmup and training 300 epochs in total. The learning rate decays at epochs 220 and 260 by a factor 0.1. The lower-level optimization use SGD with an initial learning rate 0.1, momentum 0.9, and weight decay $1e - 4$, over 300 epochs. At the same time, the upper-level hyper-parameter optimization also uses SGD with an initial learning rate 0.05, momentum 0.9, and weight decay $1e - 4$. To get better results, we initialize $l$ using LA loss in experiments in Table 2. For a fair comparison, there is no initialization in other experiments. For LA and CDT results in Table 2, we do grid search on the imbalanced validation dataset and retrain for a fair comparison.

For most of the experiments, except $\text{Err}_{\text{weighted}}$ in Table 3, we plot or report the average result of 3 runs. For $\text{Err}_{\text{weighted}}$ where the target weight $\alpha$ was generated randomly, we repeat ten times with different random seeds. We report the average result of 10 trails of different $\omega^{\text{test}}$ for $\text{Err}_{\text{weighted}}$. At each trial, weights $\omega_k^{\text{test}}$ are generated i.i.d. from the uniform distribution over $[0, 1]$ and then normalized.

All the experiments are run with 2 GeForce RTX 2080Ti GPUs. 

