# OpenReview forum: "Class-attribute Priors: Adapting Optimization to Heterogeneity and Fairness Objective"
_automl.cc/AutoML/2023/Conference — AutoML 2023 Workshop_

### Official Review · Reviewer_2WMX · 2023-04-10

**Potential Impact On The Field Of Automl Rating:** 4
**Technical Quality And Correctness Rating:** 3
**Clarity Rating:** 4

**Summary Of Contributions:**

The paper proposed new meta-strategy class-attribute priors (CAP) that support improving fairness objectives while adapting to the different heterogeneities of the individual classes. Results show that using CAP can lead to an improvement in fairness objectives performance.

**Actions Required To Increase Overall Recommendation:**

A couple of statements attempt to connect the benefits of CAP back to the problem with prior techniques.

**Clarity:**

The paper has been well-written and logically organized. The figures and tables are clear and well-labeled. I appreciate the broader impact statements and limitations in the last section. Supplement materials support/answer some of my questions while reading the paper.



**Overall Review:**

The paper presents a compelling argument to use the CAP meta-strategy to address differences in heterogeneities across classes when optimizing for fairness objectives.

There are minor gaps that can help strengthen the paper further. For example, In section 1.1, the authors discuss how the bilevel problem is time-consuming. However, it needs to be clarified if the proposed method is time-consuming. If yes, by what scale? It is vital to connect the benefits of CAP back to the problem with prior techniques.


**Potential Impact On The Field Of Automl:**

Dealing with the class imbalance and fairness objectives can significantly impact AutoML systems dealing with the class imbalance and fairness objectives can significantly impact AutoML systems.

**Reproducibility (Optional):**

The code is well organized, and README file has been provided with instructions. That being said, at code level, a little more documentation could help.

**Review Confidence:**

3: You are fairly confident in your assessment. It is possible that you did not understand some parts of the submission or that you are unfamiliar with some pieces of related work.

**Review Rating:**

8: Accept: Technically sound paper with major impact and strong evaluation, with perhaps some minor flaws.

**Review Summary:**

Class label attributes can influence the model performance and fairness objective when working with real-world data. While there are several methods out there in the field, there is no one method that solves them all. Different techniques can help in different instances. So I believe this CAP strategy can bolster this filed of research.

**Technical Quality And Correctness:**

Focusing on individual classes and creating a functional mapping between attributes and hyperparameters of the individual class is a very interesting approach. The equations corresponding to optimizing the function mapping A2H over validation loss and the process of parametrizing with a weight matrix have been clearly described.

I would like to ask the authors some clarifying questions.

- Was the validation set stratified? It is essential to preserve the distribution of the validation set across different runs to ensure results are consistent.
- Can authors ensure that the improvement, as expressed by error difference, accounts for the imbalance in the validation set?
- Is there a reason why the authors did not include a testing set?

---

> ### Author Response · Authors · 2023-05-02
> **Response to Reviewer 2WMX**
>
> (1) **Re: Validation set distribution questions.**
> The validation set was chosen in accordance with prior literature. During the search phase for bilevel CAP and post-hoc transferability experiments (Sec. 4.1), we split the training set into 80\% training and 20\% validation to obtain the optimal loss function design. For all other post-hoc experiments (Sec. 4.2 and 4.3), we follow the setup of [11,25] by training a model on the entire training dataset as the pre-train model and optimizing a logit adjustment on a balanced validation dataset. The test dataset was left untouched as it would be impractical to optimize model parameters using this dataset. Since all experiments are conducted on the same validation set, our comparison is fair across all baseline methods and CAP. We further provided post-hoc experiments with imbalanced validation in Table 6.
>
> [11] Moritz Hardt, Eric Price, and Nathan Srebro. Equality of opportunity in supervised learning. arXiv preprint arXiv:1610.02413, 2016.
> [25] Aditya Krishna Menon, Sadeep Jayasumana, Ankit Singh Rawat, Himanshu Jain, Andreas Veit, and Sanjiv Kumar. Long-tail learning via logit adjustment. arXiv preprint arXiv:2007.07314, 2020.
>
> (2) **Re: If the proposed method is also time-consuming?**
> Bilevel optimization can be time-consuming, primarily due to the computation of the Hessian-vector-product. However, this can be bypassed by employing post-hoc logit adjustment. In this work, we focus on the development of a loss function through bilevel optimization and post-hoc logit adjustment. Additionally, we explore the interconnection between bilevel CAP and post-hoc optimization by transferring obtained hyper-parameters. Employing post-hoc logit adjustment or the construction of a loss function via post-hoc optimization has the potential to significantly enhance the efficiency of the process by approximately 5x. The details of transferability are shown in Appendix C.
> As reported in reference [23], bilevel optimization can be unstable and difficult to optimize, particularly when optimizing the inner loss function. The authors propose a lengthy warm-up phase, which can be time-consuming. In Appendix B, we discuss how the use of Class-Attribute Prioritization (CAP) can improve training stability and eliminate the need for a prolonged warm-up phase. Our proposed $\text{CAP}_{\text{Bilevel}}$ method, which incorporates attributes, not only leverages prior knowledge but also improves optimization stability by simultaneously updating weights for all classes through the dictionary design.  CAP with bilevel optimization achieves a classification error rate of 20.16 on CIFAR10-LT and 56.55 on CIFAR100-LT with only five epochs of warm-up, representing an improvement in both computational efficiency and test performance.
>
> [23] Mingchen Li, Xuechen Zhang, Christos Thrampoulidis, Jiasi Chen, and Samet Oymak. Autobalance: Optimized loss functions for imbalanced data. In A. Beygelzimer, Y. Dauphin, P. Liang, and J. Wortman Vaughan, editors, Advances in Neural Information Processing Systems, 2021.

---

### Review · Reproducibility_Reviewer_9zhG · 2023-04-11

**Completeness Of Code And Dataset Supplement Rating:** 3
**Usability And Ease Of Reproducibility Rating:** 3

**Actions Required To Increase The Reproducibility And Overall Recommendation:**

- Include a requirements.txt to list necessary dependencies
- Provide clear instructions on which configuration file to use to reproduce each figure
- Ensure the code runs without errors and includes missing function definitions
- Consider including a Docker or Singularity file to ensure consistent and reproducible environments

**Completeness Of Code And Dataset Supplement:**

The authors provide code and instructions to run it via a readme file. They further provide several configuration files to run different versions of the algorithm. Unfortunately, it is not clear what specific instantiation is run with one particular configuration file. Furthermore, no code seems to be provided to generate the figures from the paper.

**Overall Reproducibility Review:**

The paper proposes a method called class-attribute priors (CAP) for personalizing the optimization strategy of individual classes in classification problems. The authors evaluate CAP on various datasets, demonstrating its ability to adapt to different types of heterogeneities. The authors provide code and instructions to run it but lack specific information about which configuration file to reproduce some figures from the paper. The code also lacks a requirements file, and some functions are not defined, indicating that some files were not included in the submission. The code comments could be improved.

**Review Confidence:**

3: You are fairly confident in your assessment. It is possible that you did not understand some parts of the submission or that you are unfamiliar with some pieces of the code or data.

**Review Rating:**

4: Weak Reject, you were not able to reproduce some critical aspects of the paper, but believe it is likely possible with additional effort.

**Review Summary:**

While the proposed method, CAP, is promising for optimization strategies in classification problems with heterogeneous class labels, the lack of specific information on reproducing some figures and the incomplete code submission raises concerns about the reproducibility of the results. Additionally, the lack of a requirements file and issues with some functions further complicate the reproducibility of the paper. Therefore, more effort is needed to improve the reproducibility of the paper before it can be recommended for wider adoption.

**Summary Of Necessary Code And Dataset Supplement:**

This paper proposes a method to personalize the optimization strategy of individual classes in classification problems that exhibit heterogeneities across classes. The proposed meta-strategy, called class-attribute priors (CAP), generates a class-specific strategy based on attributes of that class, leading to improvements over the naive approach of assigning separate hyperparameters for each class.

The authors evaluate CAP on various datasets, showcasing its ability to adapt jointly to different types of heterogeneities.


**Usability And Ease Of Reproducibility:**

It was unclear to me how to reproduce the results from the paper. The authors provide a large set of configuration files, but it remains unclear which configuration file is required to reproduce some figures from the paper. Furthermore, the code does not initially execute without errors:

* "from utils.metrics import print_num_params" breaks, function seems to be defined nowhere
* "from utils.metrics import topk_corrects" breaks, function seems to be defined nowhere

It seems like some file was not included in the submission.

---

> ### Author Response · Authors · 2023-05-01
> **Response to Reproducibility Reviewer 9zhG**
>
> We thank you for your insightful comments and suggestions. Sorry for the inconvenience. We missed the utils folder for the previous version and cause errors. We update our code based on your suggestion. We upload the new anonymized code with https://anon-github.automl.cc/r/auto_fair-3BCE/README.md
>
> ### Include a requirements.txt to list necessary dependencies
>
> We appreciate your recommendation. As per your suggestion, we have add a requirements.txt file and also produced an environment.yml file that can be utilized to establish the requisite environment. For further information, kindly refer to the Installation section in the README.md file.
>
> ### Provide clear instructions on which configuration file to use to reproduce each figure
>
> We add a folder called configs_new containing the configuration files. The detail instructions are explained as examples as README.md.
>
> To reproduce the post-hoc Cifar10 results in Table 2. Run  ```configs_new/cifar10/train_cifar10_posthoc.sh ```
>
> To reproduce the bilevel Cifar10 results in Table 2. Run  ```configs_new/cifar10/train_cifar10_bilevel.sh ```
>
> To reproduce the results in Table 3 run  ```configs_new/cifar100/objective.sh ```
>
> To reproduce the results in Figure 2 run ``` configs_new/cifar100/objective_figure.sh```
>
> Presently, we collate the outcomes and manually generate the figures. Subsequently, we plan to add a separate code file to incorporate result aggregation and figure production.
>
> ###  Ensure the code runs without errors and includes missing function definitions
>
> We sincerely apologize for any inconvenience caused. The prior version did not include the "utils" folder, leading to errors. We have rectified this issue by adding the necessary files.
>
> ### Consider including a Docker or Singularity file to ensure consistent and reproducible environments
>
> To ensure consistent and reproducible environments, we produced an environment.yml file that can be utilized to establish the requisite environment. The environment could be created by using
>
> ```conda env create -f environment.yml```

---

### Official Review · Reviewer_mnow · 2023-04-12

**Potential Impact On The Field Of Automl Rating:** 4
**Technical Quality And Correctness Rating:** 4
**Clarity Rating:** 4

**Summary Of Contributions:**

This paper focuses on the supervised classification problem where the number of classes are relatively high, and the classes are significantly imbalanced, and the goal is to enable learning of a fair classifier with explicit focus on the performance of the classifier on the smaller "tail classes". To this end, the paper focuses on the use of class-specific hyperparameters that allow for fair classification, and proposes the use of "class attributes" -- class-specific data meta-features -- to learn a mapping from the class-attributes to class-specific hyperparameters. This setup is well motivated since the number of hyperparameters to be optimized in the vanilla case can be high for large number of classes, and the use of class-attributes and a corresponding mapping significantly reduces the number of parameters to be estimated. This is especially useful when considering classes with very few examples. The paper demonstrates how some of the existing schemes to address this problem can be obtained as special cases of the proposed class-attribute based scheme. The empirical evaluation demonstrates that the proposed use of class-attributes leads to significant performance improvement on tail classes across multiple datasets, fairness objectives, and class heterogeneity, relative to existing schemes.


**Actions Required To Increase Overall Recommendation:**

I am already scoring this paper fairly high. I will increase my score if the authors can improve my understanding of this paper and highlight the challenges, if any, in learning the **A2H** mapping and how they mitigate them.


**Clarity:**

The paper is really well written and clearly articulates its contributions. The introduction clearly explains the problem of tail-classes and class-fair learning. The literature review does a great job at positioning the proposed scheme against existing approaches. Throughout the paper, the proposed scheme is well motivated, and the authors also highlight (wherever appropriate) how the proposed scheme is technically similar or different than the existing schemes.

However, there are some technical details that can be clarified better:

- The paper discusses the "transfer" between "bilevel loss function design" and "post-hoc optimization". However, the discussion in Lines 209-214 is not very clear to me. Some technical details are missing, and maybe this part requires prior knowledge of the post-hoc optimization of [25]. It would be good to provide details somewhere regarding how the $\Delta'$ and $\mathbf{l}'$ are computed, and why we use $\Delta_k'^{-1}$ in **A2H'** but $\Delta_k$ in general **A2H**.

- It is not also clear if attributes like "class conditional error" $\mathcal A_{\text{DIFF}}$ are something that are pre-computed or something that are maintained during the learning and hence are progressively changing as the optimization algorithm progresses. If they are precomputed, what models are used to compute this error?

- In the bilevel formulation (between lines 194-195), is the $\mathcal L_{\text{train}}$ the parametric cross-entropy function and the $\mathcal L_{\text{val}}$ the different fairness objectives such as balanced error, quantile error or CVaR? Is that how the proposed framework can work with different fairness objectives? If this is not the case, what are the different objectives in the bilevel formulation, and how are the different fairness objectives optimized for in the experiments in Section 4.2?


**Overall Review:**

To the best of my understanding, this paper studies an important problem of fair classification, and presents a fundamental framework for class-fair learning with significant practical impact on tail classes. The paper is well written, with the contributions well articulated, the proposed approach well positioned against existing work. The empirical evaluation clearly motivates each experiment, highlighting different properties of the proposed scheme, and explicitly discusses the conclusion from each of them. It is also relevant to AutoML in the sense that it is trying to configure hyperparameters (such as the hyperparameters of the loss function) and making use of data meta-features.


I am unable to identify any significant negative aspect of this paper. There are some minor clarification issues (as I have discussed in the **Clarity** section, but nothing significant that cannot be easily fixed.


**Potential Impact On The Field Of Automl:**

This paper provides the performance improvements for classifiers on tail-classes. This is of importance to the general area of machine learning beyond just AutoML. Given various problems with large number of classes, this work can have significant practical impact.


**Review Confidence:**

4: You are confident in your assessment, but not absolutely certain. It is unlikely, but not impossible, that you did not understand some parts of the submission or that you are unfamiliar with some pieces of related work.

**Review Rating:**

9: Strong Accept: Technically flawless paper with major impact and strong evaluation, with no obvious flaws. Should be highlighted in the program.

**Review Summary:**

I think this paper studies an important problem and presents a well-motivated intuitive approach with strong empirical performance against existing baselines. I do not recognize any significant negative aspect of this paper. Hence, I am recommending this paper be accepted.


**Technical Quality And Correctness:**

The paper proposes a very intuitive and clever approach of utilizing class-attributes to reduce the number of hyperparameters that need to be learned by treating hyperparameters as functions of class attributes instead of free variables since that would allow us to model the per-class hyperparameter dependency with a smooth function. The approach is a meta-algorithm, and can thus work with any model, class attributes, and class-fair objectives. The authors have motivated their choices in the paper really well.


The empirical evaluation clearly and thoroughly demonstrates the capabilities of the proposed scheme to (i) improve upon existing schemes, (ii) be able to work with any fairness objective. The empirical evaluation also highlights the advantage of using different class attributes with an ablation study.

---

> ### Author Response · Authors · 2023-05-02
> **Response to Reviewer mnow**
>
> (1) **details of the "transfer" between "bilevel loss function design" and "post-hoc optimization".**
>
> First, intuitively, in convex optimization, as there is only a global optimum, when we offset the logits in the loss function by $\Delta$ and $l$, the new logits will also be scaled and offset by $\frac{1}{\Delta}$ and $-l$ in comparison to the original logits. In this case, the transferability between loss function design and post-hoc optimization will always be true in convex problems. However, in non-convex optimization, the loss function design and post-hoc adjustment may lead to different local minima. Despite this, our experiments have demonstrated that transferability still exists between loss function design and post-hoc optimization even in non-convex scenarios.
>
> Our main idea of "transfer" between "bilevel loss function design" and "post-hoc optimization" is inspired by logit-adjustment[25], where they provide detailed explanation and justification section 5.1. The logit adjusted loss directly enforce the class prior offset while learning the logits, rather than doing this post-hoc. Modified loss with $\iota$ is equivalent to using a scorer of the form $g_y(x)=\Delta f_y(x)+\iota$. We thus have $argmax_{y \in[K]} f_y(x)=argmax_{y \in[K]} \frac{1}{\Delta}(g_y(x)-\iota)$. Consequently, one can equivalently view learning with this loss as learning a standard scorer g(x), and post-hoc adjusting its logits to make a prediction. Therefore, we utilize opposite signs.
>
> [25] Aditya Krishna Menon, Sadeep Jayasumana, Ankit Singh Rawat, Himanshu Jain, Andreas Veit, and Sanjiv Kumar. Long-tail learning via logit adjustment. arXiv preprint arXiv:2007.07314, 2020
>
> (2) **Detail of generating attributes like "class conditional error"**
>
> We have included Fig.~2 to provide a detailed explanation of the algorithm. The class attributes are calculated before training. For instance, test-time class weights are determined based on the target fairness objectives and do not require any optimization. The class-conditional error, utilized as $\mathcal{A}_{DIFF}$, can be computed using the classification outcome of any pre-trained model. In our experiments, considering the computational cost, we computed the class-conditional error after pre-training for post-hoc optimization and at the end of the warm-up phase for bilevel training.
>
> (3) **How the proposed framework can work with different fairness objectives?**
>
> First, in the bilevel formulation, $\mathcal{L}_{val}$ corresponds to the fairness objectives. In our current experiments, we only optimized different fairness objectives using post-hoc optimization, with the loss function serving as the fairness objectives. This is because post-hoc optimization is a more flexible and efficient solution that enables rapid adaptation to various objectives, which aligns with our goal for quick computation. The hyper-parameters obtained from post-hoc optimization also have the potential for transferring to loss function design. We are happy to run bilevel optimization with multiple objectives if necessary. In this case, the validation loss function should be set to different fairness objectives.
>
> We further add Fig. 4 in Appendix B to demonstrate our entire training process.

---

### Official Review · Reviewer_MKuD · 2023-04-13

**Potential Impact On The Field Of Automl Rating:** 2
**Technical Quality And Correctness Rating:** 2
**Clarity Rating:** 2
**Ethics And Accessibility Rating:** Yes, regarding discrimination / bias …

**Summary Of Contributions:**

This paper proposes a novel method for adjusting the hyperparameters of a loss function to account for heterogeneity across classes in (multi-class) classification tasks. The method works by defining a set of per-class attributes, applying a set of transformations to the attributes and then learning a linear mapping from the transformed attributes to the hyperparameters. They consider two strategies: one that involves a bilevel optimization and one that performs post-hoc optimization using a second function, g, applied to the trained model, f. The authors compare their method against previously proposed methods for addressing class heterogeneity and find that their method performs favorably in terms of minimizing balanced error (as well as and other metrics) using common multi-class image classification benchmark datasets.

**Actions Required To Increase Overall Recommendation:**

I am willing to increase my score, potentially to an accept-level score, if the clarity issues are addressed and the questions raised in the "Technical Quality and Correctness" section are answered.

**Clarity:**

This work is not very well-written: there are numerous typos/grammatical errors that did hinder my ability to understand the document. A non-exhaustive list includes:

- Line 15: prior art -> prior state-of-the art (?)
- Line 30: that tailored -> that are tailored
- Line 69: establishes series of algorithms -> establishes a series of algorithms (?)
- Line 71: imbalanced dataset -> imbalanced datsets
- Line 76: corresponds -> that corresponds
- Line 79: a sufficient small imbalanced validation data -> a sufficiently small imbalanced validation dataset (?)
- Line 116: objectives that aims -> objective that aim
- Line 158: how difficult to predict that class -> how difficult it is to predict that class
- Line 173: they... aggregates -> they... aggregate
- Line 174: reducing search space -> reducing the search space
- Line 178: dictionary is -> dictionaries are (?)
- Line 181: training... is equivalent to train -> training... is equivalent to training
- Line 221: utilizing attribute -> utilizing attributes
- Line 259: stand deviation -> standard deviation (?)
- Line 289: benefit of CAP -> the benefit of CAP
- Line 312: trend to -> tends to (?)

In addition to these minor yet numerous grammatical issues, there were some structural issues that I feel are worth mentioning here:

- Figure 1 is never referenced in the main body of the text so it is not clear how it relates to the rest of the document/what the reader is supposed to takeaway from it.
- \omega_k^{test} is used in Table 1 well before it is defined by Equation (3) (and frankly, the definition in Equation (3) leaves a lot to be desired)
- A slightly more minor point but the font for the acronym CAP changes in Section 5.

**Ethics Details (Optional):**

While I acknowledge this does not remotely appear to be a part of the authors' agenda, I believe that the proposed method, when used incorrectly or maliciously, could result in biased or unfair models, i.e., models that violate the fairness criteria mentioned in the work. Again, I am not holding this against the authors (and this observation is not affecting my review in any way), I just wanted to point it out as a potential concern.

**Overall Review:**

Positives
- Motivation: this work addresses an interesting and relevant problem. I think with some (substantial) improvement, a version of this work would be well-received by the community.
- Unifying framework: I enjoyed the fact that the authors demonstrated how existing methods can be viewed as instantiations of their more general method. In this regard, I think this work presents a valuable lens with which we can understand how existing methods relate to one another and how we can continue to iterate on them.

Negatives
- Writing: as noted previously, I found this work to be very poorly written and would encourage the authors to carefully check their grammar/spelling in future iterations of this work.
- Limited exploration of attribute space: throughout the work, the authors list multiple kinds of class-level attributes that could be incorporated into their method. As such it was very disappointing to see that their experiments only considered three relatively simple attributes.
- Poorly justified design choices: related to my point above, the choice of attributes considered is not well-motivated. In addition, other aspects of their proposed method are not well justified/explored e.g., the nonlinear feature map,
- Empirical evaluation: as previously noted, it is unclear which, if any, of the empirical results are statistically significant; Appendix F seems to indicate that the majority of experiments are only repeated 3 times, which is quite low in my opinion.

**Potential Impact On The Field Of Automl:**

While this work addresses an interesting and important research question, I think that given the current state of the manuscript, specifically the lack of detail in certain aspects of their method and the limited empirical evaluation, it is unlikely to have a large impact on the community.

**Review Confidence:**

2: You are willing to defend your assessment, but it is quite likely that you did not understand the central parts of the submission or that you are unfamiliar with some pieces of related work.

**Review Rating:**

3: Reject: For instance, a paper with technical flaws, weak impact, and/or weak evaluation.

**Review Summary:**

My recommendation is to reject this work in its current form due to the weak/unclear empirical evaluation, insufficient detail in the description of the method, and poor writing.

**Technical Quality And Correctness:**

Unfortunately, I have some serious concerns/questions about the technical aspects of this work, both with the method and the empirical evaluation. I have broken up my comments in this section accordingly.

Method:
- Why do we need to introduce a nonlinear feature map to the class-specific attributes? Is this just so that previously proposed methods can be described in terms of the CAP framework? An ablation study to examine the impact of this nonlinear map would have been nice to see.
- I apologize if I missed this but for the life of me, I could not figure out how the hyperparameters (\omega, \ell, \Delta) are actually being set. For the bilevel optimization strategy, it is mentioned that there is an "initial search phase" and then a reference is provided: is the search methodology detailed in the reference? If so, a brief summary should be included as it is incredibly relevant to the proposed method. Or is this initial search phase just a simple grid search, as mentioned in other places? Similarly, for the post-hoc optimization strategy, it is simply stated that g is optimized following the same setup as some references; again, I think it is important that the optimization is at least briefly described and not just deferred to a reference.

Empirical evaluation:
- How is the nonlinear map specified in Appendix F chosen? In particular, the (hyper?)hyperparameter \beta being set to 0.75 is a bit arcane: was there some sort of tuning done to determine this value?
- Does the bold in Tables 2 and 3 just indicate the best method or does it indicate statistical significance? If the latter, what statistical test was used to determine significance? From Appendix F, it seems like most experiments were only repeated 3 times, which to me feels too low to determine any sort of statistical significance...
- Relatedly, Figure 2(a) and (b) should probably have error bars; the difference between the proposed method and the baselines seems quite small and I would be curious if they are significant.
- Also from Appendix F, it is stated that the \omega_k^{test} are sampled from a uniform distribution. This seems to contradict the condition in Equation (3) that says the \omega_k^{test} sum to K: are the values normalized after sampling?
- I am curious about the computational gains associated with the proposed method relative to the baselines: the authors mention some big O computational costs that seem to favor their method but it would be nice to have some empirical evidence to see how those theoretical gains translate to practice.

---

> ### Author Response · Authors · 2023-05-02
> **Response to Reviewer MKuD**
>
> (1) **Re: Why and how to design the nonlinear function map? How to determine $\beta$ value?**
> These functions are chosen to be polylogarithms or polynomials inspired by [25, 38]. In Appendix E (Appendix D in the old version), we also provide theoretical justification for the nonlinear map. Also, thanks to the adaptability of the method, any potential functions could be used.
>
> Throughout all our experiments, including those conducted on various datasets, utilizing two optimization methods (bilevel and post-hoc), and targeting distinct fairness objectives, we employed a fixed $\beta$ value and a nonlinear map. Our results demonstrated promising outcomes, suggesting that the result is not sensitive to $\beta$ value and the nonlinear map.
>
> [25] Aditya Krishna Menon, Sadeep Jayasumana, Ankit Singh Rawat, Himanshu Jain, Andreas Veit, and Sanjiv Kumar. Long-tail learning via logit adjustment. arXiv preprint arXiv:2007.07314, 2020
>
> [38] Han-Jia Ye, Hong-You Chen, De-Chuan Zhan, and Wei-Lun Chao. Identifying and compensating for feature deviation in imbalanced deep learning. arXiv preprint arXiv:2001.01385, 2020.
>
>
> (2) **Clarfication of algorithm and how parameters are optimized**
>
>  A detailed explanation is presented in ”Response about clarity and algorithm details”. We appreciate your suggestion. As per your recommendation, we have added Appendix B, which contains Figure 4, to demonstrate our entire training process methods and clarify the algorithm.
>
> (3) **More runs of experiments.** Thank you for your suggestion. We will add more runs for each experiment for the final manuscript.
>
> (3) **Re: If the $\omega_k^{test}$ values normalized after sampling?**
>
> Yes. It needs to be normalized. We clarify it in Appendix G (Appendix F in the old version).
>
> (4) **Re: Computation gains.** First, we would like to emphasize that our big O notation focuses on the comparison of the search space with AutoBalance. AutoBalance searches in an $\mathcal{O}(K)$ space, where each class has its individual hyper-parameter to optimize. Even with the "group aggregation" strategy, which shares the same hyper-parameter for classes with a similar number of examples, the search space is only reduced by a constant, still yielding an $\mathcal{O}(K)$ search space. In contrast, within the CAP framework, $M$ is only related to the number of attributes $n$ and $|\mathcal{F}|$, making it a constant. Therefore, we state in the introduction that its order of complexity belongs to $\mathcal{O}(1)$.
>
> Furthermore, we would like to clarify that the advantages of our method are not solely attributable to the reduction in the number of parameters. Instead of optimizing the parameters individually, which can result in fragility for tail classes with limited data, our approach facilitates the linking of all classes during training through weight-sharing. Consequently, our approach improves training stability and avoids overfitting compared to other methods.
>
> Additionally, we extend our algorithm to post-hoc functions and multiple objectives, providing a flexible and efficient solution that enables rapid adaptation to various objectives.
>
> (5) **Re: typos/grammatical errors/structural issues.** Thanks for your valuable suggestions, we modified them in our new manuscript.

---

> > ### Comment · Reviewer_MKuD · 2023-05-11
> > **Thank you**
> >
> > I thank the authors for their response above. Unfortunately, some of my concerns were either unaddressed (e.g., statistical significance) or inadequately addressed (e.g., a promise of additional repetitions) so I am holding my score fixed. While I do believe that this idea has merits and I would be excited to review subsequent iterations of this work, in its current form, I do not believe it is a strong candidate for acceptance.

---

### Official Review · Reviewer_6kuv · 2023-04-13

**Potential Impact On The Field Of Automl Rating:** 2
**Technical Quality And Correctness Rating:** 3
**Clarity Rating:** 1

**Summary Of Contributions:**

Real-world datasets may have imbalanced classes (e.g., certain classes may have many examples, while others have few, leading to a long-tail in the class distribution) or may have noisy labels. These data imbalance issues pose challenges for both standard classification accuracy and fairness objectives. To address such imbalances, one may wish to treat each class differently, for example upweighting the contribution of examples from rare (e.g., under-represented) classes.

This paper proposes an approach called Class-Attribute Priors (CAP) for meta-learning a mapping from class attributes (such as the frequency of a class in the dataset, the level of label noise, similarity to other classes, and training difficulty) to an embedding vector that can be passed to a downstream algorithm to modulate the training algorithm on a per-class basis.

The authors use CAP to meta-learn the hyperparameters of the following loss function (which was used in prior work, such as AutoBalance),
$$
\ell(y, f(x)) = \omega_k \log \left( 1 + \sum_{k \neq y} e^{\ell_k - \ell_y} \cdot e^{\Delta_k f_k(x) - \Delta_y f_y(x)} \right)
$$
where $\{ \omega_k, \ell_k, \Delta_k\}$ are class-specific hyperparameters representing the overall loss weighting factor $\omega_k$, the additive logit adjustment $\ell_k$, and the multiplicative adjustment $\Delta_k$.

The $n$ attributes for class $k$, denoted $\mathcal{A}_k$, are mapped to an embedding space by a fixed, nonlinear mapping $\mathcal{F}(\mathcal{A}_k) : \mathbb{R}^n \to \mathbb{R}^M$ , and then this embedding is mapped to a vector of hyperparameters $\mathcal{S}_k = W \mathcal{F}(\mathcal{A}_k)$ via a weight matrix $W$ that is learned. These hyperparameters are the $(\omega_k, \ell_k, \Delta_k)$ that define the loss function for the given class $k$. Conceptually, the embedding serves as a bottleneck that can map classes with similar attributes close to each other, such that the resulting class-specific losses are similar. The idea is that the relatively small number of class attributes can reduce the number of meta-parameters that need to be learned, e.g., $W$ has dimension $s \times M$ rather than $s \times K$ as would be the case if learning separate hyperparameters for each of the $K$ classes directly.

The authors propose two variants of the approach: in the first, they formulate the learning of $W$ as a bilevel optimization problem, where the inner problem optimizes model parameters using the current parametric loss function, and the outer problem evaluates the trained parameters on a validation set, targeting a fairness objective. The second approach performs post-hoc optimization of a function $g$ that modifies the logits output by $f(x)$ to minimize the loss on a test dataset (without performing bilevel optimization).

The paper evaluates CAP on long-tail variants of CIFAR and Imagenet. They use CAP to optimize for different fairness objectives, including quantile class performance and conditional value at risk (CVaR), and they compare to a few prior approaches for dealing with class imbalance. They also measure balanced error, and check which attributes are most useful on different tasks in an ablation study.

**Actions Required To Increase Overall Recommendation:**

In order for this paper to meet the bar for acceptance, I believe it would have to be rewritten to improve clarity, and remove obfuscation. In addition, please see the comments regarding ablations and the comparison to AutoBalance listed in the Overall Review box.

**Clarity:**

The paper has serious clarity issues, discussed in the Overall Review box. Overall, it is hard to parse due to overly vague and abstract terminology, that constantly obfuscates rather than clarifies the main ideas.

**Overall Review:**

**Pros**

* The CAP method is simple, and yields few hyperparameters.

* The empirical results are good; CAP outperforms logit adjustment (LA), CDT, and AutoBalance on balanced error in Table 2, and outperforms LA when optimizing for two different fairness objectives in Figure 2.

* The authors implement two instantiations of CAP: 1) using CAP to learn a parametric loss function via bilevel optimization; and 2) using CAP for post-hoc logit adjustment.


**Cons**

* Overall, the paper is not well-written, and this is a major weakness. The method is very simple, but it is not clearly explained; the writing is very wordy, and obfuscates simple ideas using overly abstract and complex terminology. I believe that the paper would benefit from a significant rewrite.

* The paper does not provide an algorithm box detailing CAP.

* I think that the method name "class-attribute priors" is not well-explained; in what way is this a prior? It does not seem to have a Bayesian interpretation. A clearer method name could be "class-attribute embeddings." Also, with regard to method naming, both A2H and CAP are used at different points in the paper, while it would be clearer to use one consistently.

* The introduction---especially the last paragraph from L64-L67---overstates the contribution of this work. To a large extent, this paper can be seen as a minor extension of AutoBalance leveraging a mapping from class attributes to embedding vectors. I believe that the authors should make this clear, rather than stating that "This work makes key contributions to fairness and heterogeneous learning problems in terms of methodology, as well as practical impact."

* The paper downplays its similarities to AutoBalance [1]. Section 3.2 of this paper ("CAP for loss function design") is nearly identical to AutoBalance, the only difference being the use of learned class embeddings. The loss function that is learned is identical to that in [1], and the bilevel optimization setup is the same. Also, part of this paper implies that the hyperparameter dimensionality for AutoBalance scales as $O(K)$ where $K$ is the number of classes; however, this scaling issue was already addressed in Section 2.2 in [1], where they use $K' < K$ dimensional hyperparameters, clustering classes together when they occur with similar frequencies in the dataset. While this is briefly mentioned in Lines 171-177 of this paper, the majority of the paper is still written in a way that suggests that reducing the hyperparameter dimensionality is a novel contribution, which is not the case.

* Overall, the novelty of the paper is limited (and overstated), as it can be interpreted as a simple extension of AutoBalance that learns embedding vectors representing classes, based on attributes of those classes.

* What meta-parameters were used to train the AutoBalance baseline? In particular, what was the dimensionality of the hyperparameter space (how many groups $g$ were used), and how does it compare with CAP?

* L150: Why is this parameterization used for $\Delta$? $\Delta = \text{sigmoid}\left( \sqrt{K} \frac{\mathcal{D} w_{\Delta}}{\| \mathcal{D} w_{\Delta} \|} \right)$

* Because the loss function parameterization is the same as AutoBalance (among other related works), did the authors of CAP try adding the personalized data augmentation (PDA) from Eq. 2.2 in [1]? This would be interesting to see with CAP, as PDA was justified theoretically in [1]. It would be useful to evaluate the impact of using class attributes for data augmentation compared to learning separate parameters for each class.

* No experimental results are provided for the iNaturalist-2018 dataset, which was used in [1].

* None of the components of Figure 1 are described in the main text or figure caption. Figure 1 is not even referenced in the main text at all. The caption should give the necessary context to parse the figure; however, the caption is vague and does not ground any of its statements in the figure. Just some of the questions that arise: What does $S$ correspond to? What does the "Function" box around the attributes box mean? Why is CAP written on the arrow connecting the function to $S$? What does the dictionary inside the attributes box refer to? What does the dashed line from the "full dataset" to the "sample size and distribution shift" plot denote? Why is "heterogeneous" color-coded the same green as the dashed arrow? How should the green and blue points be interpreted (are they two clusters of examples with labels shown by the colors, to illustrate label noise)?

  - As with the rest of the paper, the caption of Figure 1 contains overly abstract and vague language. What is the "global dataset"? What is a "composition of heterogeneous sub-datasets induced by classes"? Why is the term "strategy" used interchangeably with "hyperparameters"?

* The right-hand side of Figure 1 does not seem to be related to the LHS. The caption does not draw any connection between the RHS and LHS; I think that the information in the RHS would be better suited for a table, or simply discussed in the main text without a figure. Otherwise, the reader is left wondering which parts of the LHS are connected to each statement in the RHS.

* The term "meta-strategy" is not clearly defined.

* I think that it would be good to include more ablation studies over CAP. For example, what if only a subset of the parameters of the loss function are tuned (similarly to ablations in [1])? What happens as more features (class attributes) are passed into the embedding layer, increasing the hyperparameter dimensionality of CAP? What if the number of attributes is artificially inflated, either by adding new, random attributes, or by replicating existing attributes several times? Such investigations may be useful to check whether performance improvements are due to implicit regularization from a small number of attributes (and therefore few hyperparameters). If there are many hyperparameters, does CAP overfit to the validation set?

* In some experiments, CAP is only slightly better than AutoBalance, and it is not clear where the improvement comes from (thus the need for more ablations).

* The appendix section dedicated to experimental details is sparse.


**Minor**

* In the abstract, L12-L13, it should be clarified what type of improvement this refers to, e.g., compute/memory requirements, or performance on accuracy or fairness?

* L30: "that tailored" --> "that are tailored"

* L40: "treating hyperparameters as free variables" --> This terminology is unnecessarily confusing; "free variables" are not defined anywhere.

* L47: "as it requires $\mathcal{O}(1)$ --> Shouldn't this be $\mathcal{O}(M)$ where $M < K$?

* L259: "stand deviation" --> "standard deviation"

* L295: What does "in the context of dataset heterogeneity adoption" mean?


**Potential Impact On The Field Of Automl:**

This work is an application of meta-learning to loss function design and post-hoc logic adjustment; it does not make direct methodological contributions to the field of AutoML. However, the idea of learning a class embedding based on certain attributes (such as frequency) may be useful for other meta-learning/AutoML applications. It follows a line of work on AutoML for fairness, and may be cited by other papers in this area.

**Reproducibility (Optional):**

I appreciate that the authors have provided code with their submission. I have not run it, but I believe that the work is reproducible.

**Review Confidence:**

4: You are confident in your assessment, but not absolutely certain. It is unlikely, but not impossible, that you did not understand some parts of the submission or that you are unfamiliar with some pieces of related work.

**Review Rating:**

3: Reject: For instance, a paper with technical flaws, weak impact, and/or weak evaluation.

**Review Summary:**

While the problem is worth studying, and the results are decent, this paper needs a significant re-write to improve clarity. The clarity is a major issue.

**Technical Quality And Correctness:**

I believe that the method is technically sound, and the results are likely correct. However, the paper does not provide any theoretical justification for CAP.

---

> ### Author Response · Authors · 2023-05-02
> **Response to Reviewer 6kuv**
>
> We thank you for your insightful comments and suggestions. We have revised the paper accordingly with the changes marked in red color.
>
> (1) **Re: Not well written.**
> We express our gratitude for your valuable feedback and suggestions. We have made the revisions to the paper. The modifications have been highlighted in red for ease of identification.
>
>
> (2) **Re: Algorithm box.** We have revised the method introduction and added two new figures to further explain the algorithm. A detailed explanation is presented in "Response about clarity and algorithm details".
>
> (3) **Re: Method name and definition of "prior".** We appreciate the feedback regarding the term "prior" and would like to clarify that in our technique, "prior" has a broader definition that encompasses Bayesian priors. We refer to the priors as Class-attribute Priors because the attributes are computed prior to training, thereby constituting prior information.
>
> (4) **Re: Difference between CAP and A2H.** To avoid confusion, we would like to emphasize that A2H and CAP are not identical. CAP refers to the complete framework, which comprises the class attributes design, A2H, and training via either Bilevel optimization or post-hoc optimization. On the other hand, A2H is responsible for mapping the attributes to a class-specific strategy. We have added Fig.~2 to provide a more detailed explanation of the approach.
>
> (5) **Re: Novelty, contribution and relation to AutoBalance.** First, our work is not merely an additive extension of AutoBalance but instead proposes several different contributions. AutoBalance focuses on bilevel optimization and aims to optimize balanced accuracy. Their method directly optimizes the final weights $\mathbf{\omega}$ and $\mathbf{l}$, resulting in a large search space, and they have to group tail classes for better stability when the number of validation examples is extremely small. In contrast, CAP tackles multiple fairness objectives, incorporates post-hoc optimization, and utilizes class attributes to develop a more comprehensive solution. These aspects go beyond the scope of AutoBalance and demonstrate the novelty of our approach.
>
> Then, regarding the relationship with AutoBalance, we do not intend to downplay the similarities with AutoBalance, and we acknowledge its importance in our implementation. However, it is crucial to note that our work also incorporates post-hoc optimization, and multiple fairness objectives, which distinguishes our approach from AutoBalance.
>
> (6) **Re: meta-parameter of AutoBalance baseline.** We employed the same parameters as AutoBalance when reproducing their results. In the case of CIFAR10-LT, CIFAR100-LT, and ImageNetLT datasets, we utilized 1,10, and 20 classes per cluster, respectively.
>
> (7) **Re: Why choose $\Delta={sigmoid}(\sqrt{K}\frac{\mathcal{D}\omega_{\Delta}}{\|\mathcal{D}\omega_{\Delta}\|})$**
> First, we normalize $\mathcal{D}\omega_{\Delta}$ to make it unit length and scale it back to $\sqrt{K}$ length to maintain the magnitude of the vector. That is a common practice in machine learning and data processing to ensure that the features of the data are on the same scale. We also use a sigmoid function to normalize the delta value following the AutoBalance formular. As detailed in Appendix C, without proper early stopping or regularization, the delta value can increase continuously, leading to a stretched logits distribution where the logits become progressively larger.
>
> (8) **Re: Experiment with Personalized Data Augmentation (PDA) and iNaturalist.** In our paper, our primary focus is on fairness objectives and post-hoc adjustment. Our post-hoc adjustment method offers a flexible and efficient solution that enables rapid adaptation to various objectives. In contrast, PDA demands higher complexity and computational time, which is why we did not incorporate it in our current experiments. However, if deemed necessary, we are more than willing to include additional experiments concerning PDA to further strengthen our work and address the reviewer's concern.
>
> (9) **Re: Fig 1, discussion and caption**
> We have modified the caption and added more discussions in the main context in the new manuscript.
>
> (10) **Re: Term "meta-strategy" is not clearly defined.** As our final strategy, the adjustment weights, are generated by the A2H algorithm, we refer to A2H as a meta-strategy. This is because it determines the appropriate training or adjustment strategy to use for a given dataset and fairness objective. This usage is consistent with the common understanding of the "meta-" prefix, indicating a higher level of abstraction that guides the selection of the underlying strategy. We will provide a more detailed explanation in the main context.

---

> > ### Author Response · Authors · 2023-05-02
> > **Response to Reviewer 6kuv, part 2**
> >
> > (11) **Re: More ablation studies over CAP.** Thanks for your suggestion. Though we didn’t add a separate section for ablation experiments, we provided some experiments to support our claim.
> >
> > 1) **What if only a subset of the parameters of the loss function are tuned?** It is important to note that our contribution does not pertain to the proposal of the loss function or the comparison of the parameters' roles. The family of loss functions has already been proposed, with theoretical insights provided in previous studies, particularly for bilevel optimization [25, 38, 19]. However, the relationship between loss function design and post-hoc adjustment is not entirely clear, so we have elaborated on this in Appendix C and conducted ablation experiments to provide a more comprehensive analysis. The results are shown in Table 6. And we further analyzed the performance in Appendix D.
> >
> > 2) **What happens as more features (class attributes) are passed into the embedding layer, increasing the hyperparameter dimensionality of CAP?** The results are shown in Table.4 and discussed in Section 4.3.
> >
> > 3) **What if the number of attributes is artificially inflated, either by adding new, random attributes or by replicating existing attributes several times?** We appreciate your suggestion, and we would be willing to include the experiments if deemed beneficial. However, we would like to clarify that the training may automatically down-weight random and harmful attributes which we also observed in previous experiments.
> >
> > (12) **Re: Ablation experiments on top of AutoBalance.** Our approach yields benefits through a reduction in the number of hyperparameters and the connection of all classes during training via weight-sharing, and the effectiveness of this approach is demonstrated in the experiments. We compared our technique with Autobalance, which employs a clustering strategy to reduce the number of hyperparameters, in Tables 2 and 4. This comparison highlights that simply reducing the number of hyperparameters may not be sufficient for stabilizing the training for tail classes. Our approach (CAP) is capable of providing additional assistance with tail classes while stabilizing the training process, as demonstrated in Table 4. Additionally, in Appendix B, we show that CAP enhances training stability and reduces the necessity for warm-up in contrast to Autobalance.
> >
> > (13) **Re: The appendix section dedicated to experimental details is sparse.** We add more details in the appendix. Furthermore, we provide code and reproduce configuration for this paper.
> >
> > [25] Aditya Krishna Menon, Sadeep Jayasumana, Ankit Singh Rawat, Himanshu Jain, Andreas Veit, and Sanjiv Kumar. Long-tail learning via logit adjustment. arXiv preprint arXiv:2007.07314, 2020
> >
> > [38] Han-Jia Ye, Hong-You Chen, De-Chuan Zhan, and Wei-Lun Chao. Identifying and compensating for feature deviation in imbalanced deep learning. arXiv preprint arXiv:2001.01385, 2020.
> >
> > [19] Ganesh Ramachandra Kini, Orestis Paraskevas, Samet Oymak, and Christos Thrampoulidis. Label-imbalanced and group-sensitive classification under overparameterization. accepted to the Thirty-fifth Conference on Neural Information Processing Systems (NeurIPS), 2021.

---

> > > ### Comment · Reviewer_6kuv · 2023-05-12
> > > **Thank you**
> > >
> > > I have read the other reviews and the authors' rebuttal, and I will maintain my score. My concerns regarding writing clarity were not sufficiently addressed by the rebuttal or the updated PDF. I think more substantial re-writing is needed. I appreciate that the authors added a figure attempting to explain the method, but overall, I agree with reviewers Nafb and MKuD that the paper is unclear and hard to read in its current form. I think it could be improved in a future submission cycle.

---

### Official Review · Reviewer_Nafb · 2023-04-19

**Potential Impact On The Field Of Automl Rating:** 3
**Technical Quality And Correctness Rating:** 3
**Clarity Rating:** 2
**Actions Required To Increase Overall Recommendation:** 1. Improve clarity.

**Summary Of Contributions:**

This paper proposes a meta-learning method to choose different strategies to deal with imbalance for different classes in problems with many classes.


**Clarity:**

The paper is not ready to read.

The strategies should be introduced clearly in section 2 (problem setup) and it is only in section 3.2.

Several typos or strange wording:
-page 1/line 30: "that tailored"
-p2/l69: "establishes series of algorithms"
-p2/l46-48
-p2/l69: "establishes series of algorithms"
-p3/l76: "y" (math notation)
-p3/l76: "Above methods"
(... the paper should be reviewed carefully)


**Overall Review:**

The problem is important and the approach is very interesting and obtains good results.

However, the paper is hard to read, which reduces it's usefulness and potential impact.


**Potential Impact On The Field Of Automl:**

The problem is very relevant and the use of AutoML may be important to address it, as this paper indicates.


**Review Confidence:**

3: You are fairly confident in your assessment. It is possible that you did not understand some parts of the submission or that you are unfamiliar with some pieces of related work.

**Review Rating:**

6: Borderline Leaning Accept: Technically sound paper where reasons to accept outweigh reasons to reject. Please use sparingly.

**Review Summary:**

In spite of the potential value, the effort required to improve clarity is significant.


**Technical Quality And Correctness:**

The approach is very interesting and seems generally correct, but it's not entirely clear (e.g. p5/l193-196).

The set of meta-features is limited to "heterogeneity across individual classes" (p4/l157-158). Why not other problem characteristics (e.g. error costs or relating different classes)?

Additionally, there are several problems with notation:
-p2/l75: notation is not introduced beforehand
-p3/l108: the definition is incomplete
-p4/l136: is the last term correct?
-p4/l148-149: not clear what the feature dictionary is for

---

> ### Author Response · Authors · 2023-05-02
> **Response to Reviewer Nafb**
>
> We thank you for your insightful comments and suggestions. We have revised the paper accordingly with the changes marked in red color.
>
> (1) **Re: Clarity of p5/L193-L196**. We included 2 figures and revised the paper to further increase the clarity of our approach. A detailed explanation is presented in "Response about clarity and algorithm details".
>
> (2) **Re: meta-feature is limited to "heterogeneity across individual classes".**
> Thank you for your insightful comment on the potential limitation of CAP in relating different classes. We appreciate the suggestion to incorporate an error cost or class-wise attributes to capture more kinds of data heterogeneity. The comprehensive definition of data heterogeneity encompasses a multitude of problem characteristics, including error costs. We utilize class-conditional error and label noise ratio as attributes, both of which are a type of error cost. Additionally, our method displays notable adaptability, allowing all feasible features to be employed so long as they can be quantified as class attributes. For relating classes, We would like to highlight that CAP has the potential to be adapted to incorporate a pairwise similarity matrix or noise transition matrix. By mapping pairwise class similarities through the proper A2H functions, the CAP framework can effectively utilize the class relationships and their respective similarities. This modification allows the algorithm to account for inter-class relationships.
>
> (3) **Re: p4/l136: is the last term correct?** This term is aligned with prior studies [25,38] and we also provide theoretical justification in Appendix E (Appendix D in the old version).
>
> [25] Aditya Krishna Menon, Sadeep Jayasumana, Ankit Singh Rawat, Himanshu Jain, Andreas Veit, and Sanjiv Kumar. Long-tail learning via logit adjustment. arXiv preprint arXiv:2007.07314, 2020
>
> [38] Han-Jia Ye, Hong-You Chen, De-Chuan Zhan, and Wei-Lun Chao. Identifying and compensating for feature deviation in imbalanced deep learning. arXiv preprint arXiv:2001.01385, 2020.
>
> (4) **Re: notation, typo, and wording issues.**  Thank you for your valuable feedback. Based on your suggestions, we have revised the paper and indicated the changes in our new submission.

---

### Author Response · Authors · 2023-05-02
**Response about clarity and algorithm details**

We sincerely appreciate all reviewers for taking the time to review our paper and providing valuable reviews. Firstly, we would like to address the concerns regarding the clarity of the paper and the algorithm details. We have made significant revisions to section 3 and added Figure 2 and Figure 4 to provide further clarification.

As shown in Figure 2, **CAP** is the overall framework proposed in our paper, with **A2H** being the core algorithm. \textbf{A2H} is a meta-strategy that transforms the class-attribute prior knowledge into hyper-parameter ${\mathbfcal{S}}$ for each class through a trainable matrix $\mathbf{W}$, forming a training strategy that satisfies the desired fairness objective. The left half of Figure 2 specifically illustrates how our algorithm calculates and trains the weights. In the first stage, we collect class-related information and construct an attribute table of $n \times K$ dimension. This is a general prior, which is related to the distribution of training data, the training difficulty of each class, and other factors.

Subsequently, by utilizing the constructed attribute table $\mathbfcal{A}$, the first step of \textbf{A2H} is to compute a $K\times M$ Feature Dictionary $\mathbfcal{D}=\mathbfcal{F}(\mathbfcal{A})$ by applying a set of functions $\mathbfcal{F}$. It is important to emphasize that $M << K$, and in AutoBalance[1], the number of parameters to be trained is directly related to the number of classes $K$. Even with the group aggregation, the scale remains at the $\mathcal{O}(K)$. However, in \textbf{CAP}, $M$ is only related to the number of attributes $n$ and $|\mathbfcal{F}|$, making it a constant. Therefore, we state in the introduction that its order of complexity belongs to $\mathcal{O}(1)$. Then, in the second step, the weight matrix $\mathbf{W}$ is trained through bilevel or post-hoc methods to construct the hyperparameter $\mathbfcal{S}$. We further demonstrate our entire training process in Figure 4.

[1] Mingchen Li, Xuechen Zhang, Christos Thrampoulidis, Jiasi Chen, and Samet Oymak. Autobalance: Optimized loss functions for imbalanced data. In A. Beygelzimer, Y. Dauphin, P. Liang, and J. Wortman Vaughan, editors, Advances in Neural Information Processing Systems, 2021.